# Pharmacological evidence for a metabotropic glutamate receptor heterodimer in neuronal cells

David Moreno Delgado[1], Thor C Møller[1], Jeanne Ster[1], Jesús Giraldo[2,3], Damien Maurel[1], Xavier Rovira[1], Pauline Scholler[1], Jurriaan M Zwier[4], Julie Perroy[1], Thierry Durroux[1], Eric Trinquet[4], Laurent Prezeau[1], Philippe Rondard[1], Jean-Philippe Pin[1]*

[1]Institut de Génomique Fonctionnelle (IGF), CNRS, INSERM, Univ. Montpellier, Montpellier, France; [2]Laboratory of Molecular Neuropharmacology and Bioinformatics, Institut de Neurociències and Unitat de Bioestadística, Universitat Autònoma de Barcelona, Bellaterra, Spain; [3]Network Biomedical Research Center on Mental Health, Madrid, Spain; [4]CisBio Bioassays, Codolet, France

**Abstract** Metabotropic glutamate receptors (mGluRs) are mandatory dimers playing important roles in regulating CNS function. Although assumed to form exclusive homodimers, 16 possible heterodimeric mGluRs have been proposed but their existence in native cells remains elusive. Here, we set up two assays to specifically identify the pharmacological properties of rat mGlu heterodimers composed of mGlu2 and 4 subunits. We used either a heterodimer-specific conformational LRET-based biosensor or a system that guarantees the cell surface targeting of the heterodimer only. We identified mGlu2-4 specific pharmacological fingerprints that were also observed in a neuronal cell line and in lateral perforant path terminals naturally expressing mGlu2 and mGlu4. These results bring strong evidence for the existence of mGlu2-4 heterodimers in native cells. In addition to reporting a general approach to characterize heterodimeric mGluRs, our study opens new avenues to understanding the pathophysiological roles of mGlu heterodimers.

*For correspondence: jppin@igf.cnrs.fr

## Introduction

G-protein-coupled receptors (GPCRs) are essential in cell-cell communication and are considered as major drug targets. Although recognized as activating G proteins in a monomeric form (*Whorton et al., 2007*), numerous studies revealed their possible association into hetero-oligomers enabling allosteric controls between receptors (*Pin et al., 2007*; *Ferré et al., 2014*). The validation of this concept in vivo remains difficult and is a matter of intense debates (*Pin et al., 2007*; *Bouvier and Hébert, 2014*; *Lambert and Javitch, 2014*). The metabotropic glutamate (mGlu) receptors are members of the class C GPCRs activated by the main excitatory neurotransmitter, glutamate. These receptors are strict dimers and have until recently only been considered as homodimers (*Romano et al., 1996*; *Kunishima et al., 2000*). However, recent studies revealed the possible existence of heterodimeric mGluRs (*Doumazane et al., 2011*; *Kammermeier, 2012*; *Yin et al., 2014*; *Niswender et al., 2016*), as observed with other class C GPCRs (*Marshall et al., 1999*; *Zhao et al., 2003*; *Pin and Bettler, 2016*). The mGluRs constitute therefore an interesting model to tackle the issue of heterodimeric GPCRs in vivo.

Among the eight mGluRs, mGlu1 and 5 (group I) are mainly postsynaptic, while mGlu2 and 3 (group II) and mGlu4, 7 and 8 (group III) are predominantly found in presynaptic terminals (*Conn and Pin, 1997*; *Niswender and Conn, 2010*). Heterologous expression studies revealed that

group-I mGluRs on one hand, and group II and III mGluRs on the other hand could form hetero-dimers (*Doumazane et al., 2011*), leading to the possible existence of 16 additional mGluRs with likely specific pharmacological and functional properties. Identifying such properties is a difficult issue to address, although one can expect that they will be essential in identifying the roles of mGlu heterodimers in vivo. What limits such studies is the presence of both homodimers and heterodimers in cells co-expressing both types of mGlu subunits (*Kammermeier, 2012*; *Yin et al., 2014*; *Niswender et al., 2016*).

Among heterodimeric mGluRs, mGlu2-4 was the most studied pair due to its important physio-logical interest and because different pharmacological tools are available (*Kammermeier, 2012*; *Yin et al., 2014*; *Niswender et al., 2016*). First, in the basal ganglia and the corticostriatal pathway, these two subunits are playing an important role in movement disorders such as Parkinson's disease (*Johnson et al., 2005*). Second, previous immunohistochemistry and in situ hybridization studies suggest that mGlu2 and 4 receptors are co-localized in several brain regions (*Neki et al., 1996*; *Bradley et al., 1999*). Accordingly, mGlu2 and mGlu4 receptors could co-immunoprecipitate in native rodent tissue (*Yin et al., 2014*). However, it is difficult to detect pharmacological activation of any heterodimer using single activation protocols due to the co-existence of both homo- and hetero-dimers. Interestingly, the co-expression of the mGlu2 and mGlu4 subunits was reported to modify the pharmacology of mGlu2 and mGlu4 agonists. In addition, amongst the positive allosteric modu-lators (PAMs) of mGlu4 receptor, only one is active at the mGlu2-4 heterodimer (*Yin et al., 2014*; *Niswender et al., 2016*). The lack of effect of some mGlu4 PAMs in modulating mGlu4-mediated inhibition of cortico-striatal terminals was then used as a first evidence for the existence of mGlu2-4 heterodimers in the brain (*Yin et al., 2014*). However, in order to discern between homodimers and heterodimers, it is essential to find their specific pharmacological signature.

In the present study, we developed two different innovative approaches to characterize the phar-macological and functional properties of mGlu2-4 heterodimers specifically, without any influence of the co-existing mGlu2 and mGlu4 homodimers. We also used an innovative lanthanide-based time-resolved FRET microscopy approach (*Faklaris et al., 2015*), to demonstrate mGlu2 and mGlu4 can form heterodimers in transfected neurons. Using our heterodimer selective assays, we identified three pharmacological fingerprints that can be used to identify mGlu2-4 heterodimers in native cells. Such fingerprints could be identified in a neuronal cell line that naturally expresses both mGlu2 and mGlu4 subunits, as well as in the lateral perforant path (LPP) terminals in the hippocampus. These data bring strong evidence for the natural formation of such heterodimeric mGluRs in brain cells. Our observation will then be useful to study the function of mGlu2-4 heterodimers in the brain, and most importantly, to set up the condition to characterize other GPCR heterodimers.

## Results

### A FRET-based sensor to identify mGlu2-4-specific fingerprints

The co-expression of both rat mGlu2 and mGlu4 subunits led to the surface expression of three types of dimers: mGlu2 and mGlu4 homodimers and mGlu2-4 heterodimers. We then set up the transfection conditions to obtain an optimal expression at the cell surface of the mGlu2-4 hetero-dimer (*Figure 1A*; *Figure 1—figure supplement 1A–C*). To that aim, we co-transfected various amounts of plasmids encoding CLIP-mGlu2 and SNAP-mGlu4, and quantified the surface expression of each dimer population measuring lanthanide resonance energy transfer (LRET) in a time-resolved manner (TR-FRET) between a long life-time donor (Lumi4-Tb) and a fluorescent acceptor covalently attached to N-terminal CLIP (*Gautier et al., 2008*) or SNAP (*Juillerat et al., 2003*) tags (*Maurel et al., 2008*; *Doumazane et al., 2011*). This approach allowed the orthogonal labeling of the subunits in any dimer combinations. The use of donor and acceptor SNAP substrates allows the specific labeling of mGlu4 homodimers with a TR-FRET pair. Similarly, the use of CLIP substrates allows the measurement of TR-FRET signal originating from mGlu2 homodimers exclusively. Eventu-ally, the use of a combination of SNAP-donor and CLIP-acceptor substrates leads to TR-FRET origi-nating from the heterodimer only, since any homodimers carry either the donor or the acceptor but not both (*Figure 1A*; *Figure 1—figure supplement 1A–C*). Under these optimized conditions (40 ng CLIP-mGlu2 and 20 ng SNAP-mGlu4), the measure of the inhibition of cAMP production revealed a partial activity of the mGlu4 agonist L-AP4 and a slight loss in potency of the mGlu2 agonist

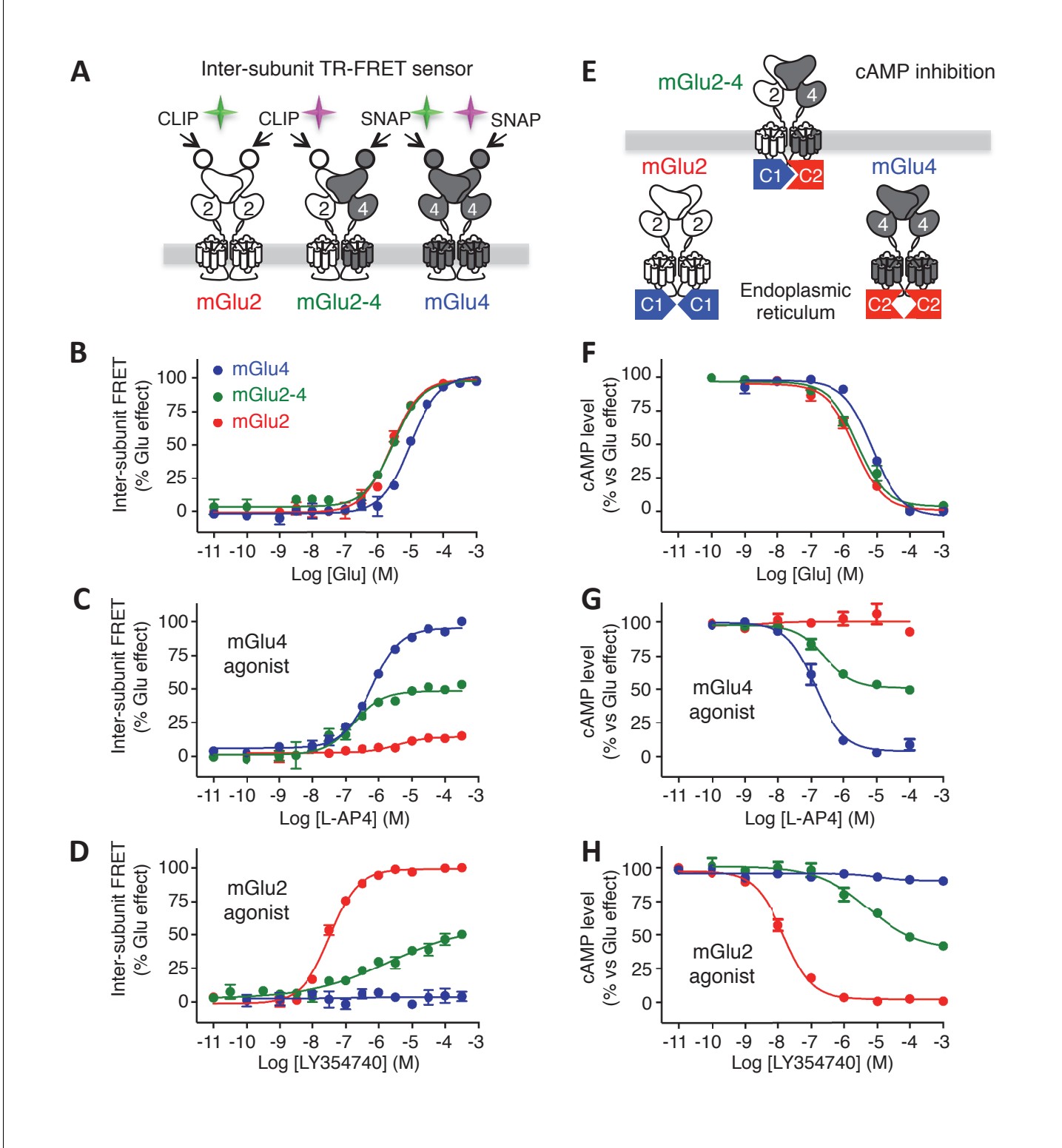

**Figure 1.** Phamacological profile of mGlu2, mGlu4 and mGlu2-4 receptors. (**A**) Schematic representation of TR-FRET mGlu sensors generated. (**B–D**) Specific effect on VFT rearrangement of the CLIP-CLIP mGlu2 (red), SNAP-CLIP mGlu2-4 (green) or SNAP-SNAP mGlu4 (blue) with increasing concentrations of the indicated compound. (**E**) Schematic representation of the C1-C2 expression control system used for a specific expression of mGlu2-4 heterodimers at the cell surface. (**F–H**) Specific detection of the inhibition of cAMP pathway using C1-C2 expression control system for mGlu2-

*Figure 1 continued*

4 (green), as well as wild-type mGlu2 (red) and wild-type mGlu4 (blue). Glutamate and the specific agonists of mGlu4 (L–AP4) and mGlu2 (LY354740) present similar pharmacological profile using both techniques. Results are mean ± SEM from three independent experiments performed in triplicates.

The following figure supplements are available for figure 1:

**Figure supplement 1.** Optimization of mGlu2, mGlu4 and mGlu2-4 expressing cells by TR-FRET and signaling.

**Figure supplement 2.** Pharmacological profile of mGlu2 (red curves), mGlu4 (blue curves) and mGlu2-4 (Green curves) expressing cells upon activation by mGlu4 or mGlu2 ligands by TR-FRET (**A–D**) and cAMP signaling (**E–H**).

**Figure supplement 3.** Validation of the use of C1-C2 constructs to get mGlu2-4 heterodimer only at the cell surface.

LY379268 (*Figure 1—figure supplement 1D–F*), as reported by others (*Yin et al., 2014*). Of note, the L-AP4 dose-response curve in cells expressing both mGlu2 and 4 subunits can be fitted with a biphasic curve (*Figure 1—figure supplement 1F*), an effect consistent with the action of L-AP4 on both mGlu2-4 heterodimers and mGlu4 homodimers. This illustrates the difficulty of analyzing the specific properties of the heterodimer under such conditions.

To examine the effect of various agonists specifically on the mGlu2-4 heterodimer at the surface of live cells, we took advantage of the large conformational change observed at the level of the extracellular domain of mGlu dimers upon activation. This conformational change led to a drastic decrease in TR-FRET signal (*Doumazane et al., 2013*) that can be followed specifically in any of the three types of dimers at the surface of cells co-transfected with CLIP-mGlu2 and SNAP-mGlu4 (*Figure 1A*). Of note, the properties of one dimer combination were then analyzed in the presence of the others.

In this assay, the glutamate potency was similar in the mGlu2-4 heterodimer and mGlu2 homodimer, higher than that on mGlu4 (*Figure 1B*), as previously reported by others (*Yin et al., 2014*). However, the potencies of the specific mGlu4 agonists were not increased in the heterodimer (*Figure 1C*; *Figure 1—figure supplement 2A,B*) suggesting no change in the affinity of mGlu4 ligands in the heterodimer. However, all of them acted as partial agonists within the mGlu2-4 heterodimer indicating that binding on the mGlu4 protomer only is not sufficient for a full activation of the heterodimer (*Figure 1C*; *Figure 1—figure supplement 2A,B*). This partial effect was more pronounced when activating mGlu4 with the partial agonist ACPT-I (*Figure 1—figure supplement 2B*). On the other hand, when activating the heterodimer with mGlu2 selective agonists, a loss in potency was observed in addition to the partial activity in the heterodimer (*Figure 1D*; *Figure 1—figure supplement 2C,D*). Interestingly, the mGlu2 agonist LY354740 displayed a strong loss in potency with a highly reduced slope ($n_H$ = 0.29) on the heterodimer (*Figure 1D*). The agonist APDC showed a right-shifted curve in the heterodimer in comparison with mGlu2 homodimer and half of the maximal response (*Figure 1—figure supplement 2C*). Of interest, DCG-IV, a high-affinity mGlu2 agonist and low-affinity mGlu4 antagonist, displayed a biphasic dose-response curve with a reduction of the response at higher concentrations (*Figure 1—figure supplement 2D*).

## Functional characterization of mGlu2-4 heterodimer confirmed the properties of LY354740

We next aimed at verifying that the pharmacological mGlu2-4 properties observed using the TR-FRET conformational sensor correlate with those measured using a functional read out. To that aim, we used a quality control system allowing the cell surface targeting of the mGlu2-4 heterodimer only (*Figure 1E*). We replaced the C-terminal tails of the SNAP-mGlu4 and CLIP-mGlu2 with a quality control system based on the modified intracellular tails of the GABA$_B$ receptor subunits (called C1 and C2) (*Brock et al., 2007*; *Huang et al., 2011*). In that situation both homodimers are retained in the endoplasmic reticulum and do not reach the cell surface, and then are not capable of generating a signal as already reported for mGlu2 (*Huang et al., 2011*), and mGlu5 (*Brock et al., 2007*) receptors. In contrast, the coiled coil interaction between the C1 and C2-tails prevents the retention of each subunit, allowing the C1-C2 heterodimer to escape from the endoplasmic reticulum and reach the cell surface (*Figure 1E*) (see (*Huang et al., 2011*) for the characterization of the mGlu2-C1 and

mGlu2-C2 constructs, and *Figure 1—figure supplement 3* for the mGlu4-C1 construct). We set up the transfection conditions to avoid even the minimum leaking of the respective homodimers that might occur during the expression of these constructs and the absence of homodimer formation was checked by TR-FRET (*Figure 1—figure supplement 3*). The inhibition of forskolin-induced cAMP by mGlu2-4 receptors revealed data that perfectly match those measured with the TR-FRET sensor assay in terms of potency, Hill coefficient and efficacy (*Figure 1F–H*, *Figure 1—figure supplement 2E–H*). The potencies of the compounds tested in these different assays are indicated in *Table 1*.

Taken together, these data revealed a low potency and low Hill coefficient for LY354740, which can be used as a first fingerprint of the mGlu2-4 heterodimer.

## Activation of both subunits in mGlu2-4 receptor is required for full activity

To examine the role of each binding site in the activation of an mGlu heterodimer, we examined the effect of mutating either site. The substitution of the conserved Tyr and Asp by Ala in the glutamate-binding site of mGlu receptors (position 216 and 295 in mGlu2), which are called YADA mutants (*Kniazeff et al., 2004*; *Brock et al., 2007*; *Doumazane et al., 2013*), strongly impairs the binding of agonists. Accordingly, mGlu2$^{YADA}$ and mGlu4$^{YADA}$ homodimers, as well as the mGlu2$^{YADA}$-4$^{YADA}$ heterodimer could not be activated by glutamate (*Figure 2*), despite their normal expression at the cell surface (*Figure 2—figure supplement 1B,C*). However, when a single subunit per heterodimer is mutated, glutamate maximal FRET change was about half the maximal response of the wild-type receptor (*Figure 2B*), consistent with a full activation requesting both binding sites occupied. As expected, no effect of the selective mGlu2 agonists could be observed in the mGlu2$^{YADA}$-4 heterodimer (*Figure 2D–F*). Similarly, selective mGlu4 agonists had no effect on the heterodimer mutated in the mGlu4 subunit (mGlu2-4$^{YADA}$) (*Figure 2C*, *Figure 2—figure supplement 1A*), but mGlu2 selective agonists retained their activity (*Figure 2D–F*) on this mutant heterodimer. Of note, the Hill coefficient of LY354740 was increased to $n_H = 0.89$ (*Figure 2E*), and the DCG-IV dose response curve was no longer biphasic, the decreased response obtained at higher

**Table 1.** Potencies (pEC50) of the indicated compound on mGlu2, mGlu2-4 and mGlu4 as determined using the TR-FRET-based conformational assay depicted in *Figure 1*, or the cAMP assay as depicted in *Figure 2*.
Data are means ± SEM of at least three experiments performed in triplicates.

| | Receptor | | | | | |
| | mGlu2 | | mGlu2-4 | | mGlu4 | |
| Compound | pEC$_{50}$ | Emax (%) | pEC$_{50}$ | Emax (%) | pEC$_{50}$ | Emax (%) |
|---|---|---|---|---|---|---|
| TR-FRET conformational sensor assay | | | | | | |
| Glutamate | 5.5 ± 0.04 | 100 ± 1 | 5.5 ± 0.04 | 100 ± 2 | 5.0 ± 0.05 | 100 ± 2 |
| LY354740 | 7.5 ± 0.04 | 99 ± 1 | 5.9 ± 0.4 | 59 ± 9 | – | 3 ± 1 |
| APDC | 5.4 ± 0.07 | 97 ± 3 | 4.8 ± 0.1 | 56 ± 3 | – | 10 ± 2 |
| DCGIV | 6.6 ± 0.06 | 79 ± 1 | 6.8 ± 0.2 | 27 ± 2 | 4.7 ± 0.2 | 16 ± 2 |
| L-AP4 | – | 14 ± 1 | 6.7 ± 0.1 | 48 ± 2 | 6.2 ± 0.1 | 95 ± 1 |
| LSP4-2022 | – | 3 ± 1 | 6.8 ± 0.1 | 53 ± 2 | 6.2 ± 0.05 | 91 ± 1 |
| ACPT-I | – | 7 ± 1 | 5.3 ± 0.2 | 26 ± 2 | 5.1 ± 0.1 | 74 ± 2 |
| cAMP assay | | | | | | |
| Glutamate | 5.6 ± 0.06 | 100 ± 1 | 5.6 ± 0.07 | 100 ± 2 | 5.1 ± 0.06 | 100 ± 2 |
| LY354740 | 7.8 ± 0.04 | 97 ± 1 | 5.3 ± 0.2 | 59 ± 6 | – | 5 ± 1 |
| APDC | 6.3 ± 0.06 | 96 ± 2 | 6.0 ± 0.1 | 50 ± 2 | – | 9 ± 3 |
| DCGIV | 6.9 ± 0.09 | 69 ± 2 | 6.6 ± 0.2 | 42 ± 3 | 5.5 ± 0.4 | −17 ± 2 |
| L-AP4 | – | 3 ± 2 | 6.6 ± 0.1 | 51 ± 1 | 6.9 ± 0.07 | 99 ± 2 |
| LSP4-2022 | – | 3 ± 1 | 6.9 ± 0.1 | 48 ± 2 | 6.9 ± 0.04 | 99 ± 1 |
| ACPT-I | – | 13 ± 7 | 5.9 ± 0.1 | 32 ± 1 | 5.7 ± 0.1 | 68 ± 3 |

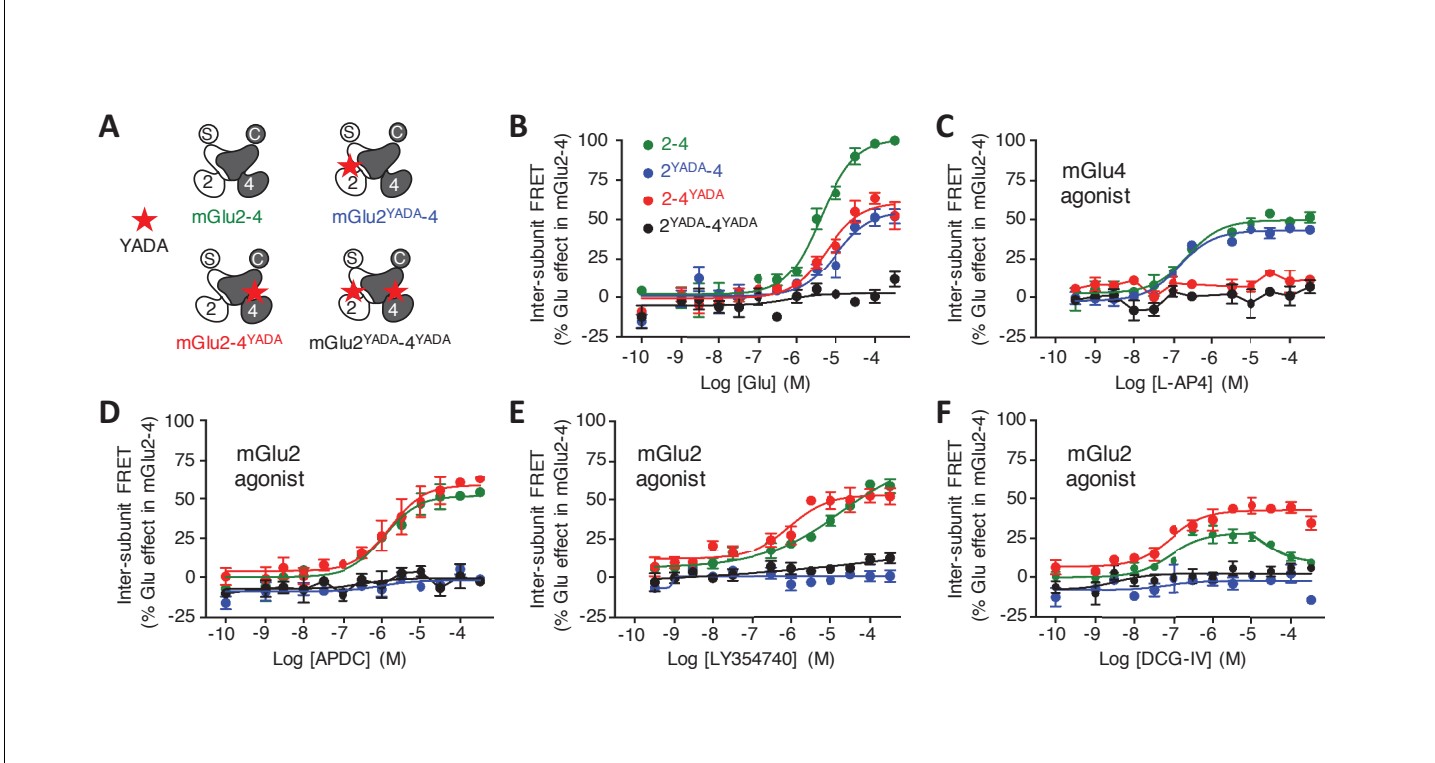

**Figure 2.** Role of each binding site in agonist-induced activity of mGlu2-4. (**A**) Schematic representation of the mGlu2-4 mutants; wild type (green), YADA mutation in mGlu2 (blue), mGlu4 (red) or both (black). (**B–F**) Effect of increasing concentrations of the indicated ligands on the mGlu2-4 TR-FRET sensor. Results are mean± SEM of three independent experiments performed in triplicates.
The following figure supplement is available for figure 2:

**Figure supplement 1.** Lack of activation of mGlu2 and mGlu4 mutated in their binding site.

dose not being observed in this mutated heterodimer (*Figure 2F*). These findings demonstrate the importance of an intact mGlu4-binding site in the complex pharmacological effect of the mGlu2 agonists LY354740 and DCG-IV on the mGlu2-4 heterodimer.

## Cooperativity between the agonist binding sites in the mGlu2-4 heterodimer

The above data prompted us to examine the influence of agonist binding in one subunit on the effect mediated through agonist binding in the second subunit. We observed that agonist binding in mGlu4 receptor increased the potency of mGlu2-specific ligands on the mGlu2-4 heterodimer (*Figure 3A–C*, *Figure 3—figure supplement 1*). In the case of LY354740, not only mGlu4 agonists increased its potency, but also restored an nH close to unity, as observed with the TR-FRET conformational sensor (*Figure 3A*) and cAMP assays (*Figure 3B,D*). These results revealed a crosstalk between mGlu4 and mGlu2 protomers within the heterodimer, an effect that can be observed both at the level of the ECDs as revealed by the TR-FRET sensor, and at the G-protein coupling site.

To quantitatively analyze the effect of mGlu4 agonists on the nH of LY354740 between ligands in the mGlu2-4 heterodimer, mathematical models were developed (*Figure 3—figure supplement 2A, B*). Assuming that mGlu2 and mGlu4 ligands bind to their respective protomers exclusively (Model 1), the mechanistic model collapses to an empirical model that can be expressed as a Hill equation with a Hill coefficient of 1 (see Appendix 1). This is not consistent with the flat slope curves displayed by LY354740 and the cooperativity exerted by mGlu4 ligands. In a second model, we then assumed that LY354740 could bind to mGlu4 VFT, with a very low affinity in mGlu4 homodimers, but with a higher affinity in the mGlu4 subunit of the mGlu2-4 heterodimer due to its binding on mGlu2 VFT.

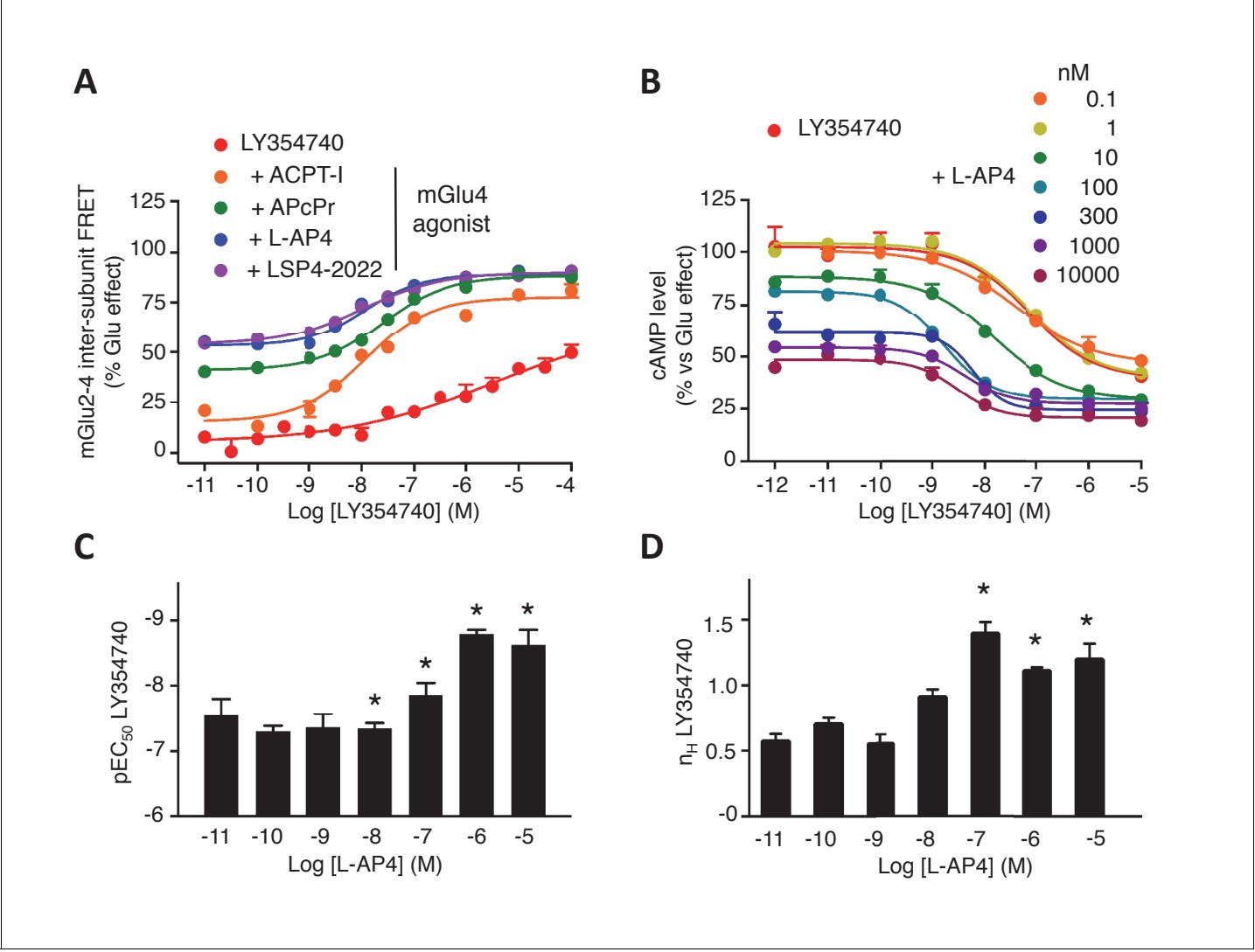

**Figure 3.** Synergistic action of mGlu2 and mGlu4 agonists in mGlu2-4 heterodimer. (**A–B**) Dose response curves of the mGlu2 ligand LY354740 in absence or presence of mGlu4 ligands (ACPT-I 10 μM, APcPr 3 μM, L-AP4 3 μM or LSP4-2022 3 μM) on the TR-FRET assay (A) and the inhibition of forskolin-induced cAMP production (B). (**C–D**) LY35740 EC$_{50}$ (C) or Hill slope (D) in the presence of the indicated concentration of L-AP4. Results are mean ± SEM of three independent experiments performed in triplicates. Curve fitting was performed by using nonlinear regression. p-values<0.05 were considered statistically significant (*).

The following figure supplements are available for figure 3:

**Figure supplement 1.** Increase in mGlu2 ligands potency in presence of mGlu4 agonist.

**Figure supplement 2.** Models for the action of agonists at the mGlu2-4 VFT domains.

For a best fitting of the LY354740 curve (*Figure 3—figure supplement 2C*), we had to assume the closed-closed state is not fully achieved because the ligand behaves as a partial agonist. In addition, two components of the functional activity had to be set up, one related with the binding to a first VFT of the heterodimer and another one related to the binding to the second protomer. Eventually, we had to assume that LY354740 binds the mGlu4 protomer after occupying first the mGlu2 binding site. This model is then consistent with L-AP4 binding in the mGlu4 VFT increasing LY354740 potency and restoring a Hill coefficient close to unity (see Appendix 1).

The synergistic effect between mGlu2 and mGlu4 agonists constitutes a second fingerprint of the mGlu2-4 heterodimer that may be useful for the identification of mGlu2-4 in neurons.

## Allosteric modulation of mGlu2-4 heterodimer

Positive allosteric modulators (PAMs) can enhance both agonist affinity and efficacy. They can also have an intrinsic agonist activity on mGluRs (*Conn et al., 2014*). Using our TR-FRET mGluR conformational sensors, we found that the mGlu2 PAMs, BINA and LY487379 potentiate the effect of glutamate (at its $EC_{20}$) in the mGlu2 homodimer, but very weakly in the mGlu2-4 heterodimer and not at all in mGlu4 (*Figure 4A*). Regarding the mGlu4 PAMs, VU0155041 activated both mGlu4 and mGlu2-4 while VU0415374 potentiated mGlu4 homodimer mainly (*Figure 4A*), as previously reported (*Yin et al., 2014*; *Niswender et al., 2016*). Co-application of mGlu2 and mGlu4 PAMs led to specific effects on the mGlu2-4 heterodimers, depending on the PAM used. Neither BINA nor VU0415374 had any effect when applied alone on the glutamate $EC_{20}$ mediated response (*Figure 4B–D*). However, a clear and strong potentiation of the glutamate $EC_{20}$ response was observed when both PAMs were applied together (*Figure 4B–D*). This synergistic effect was not observed with another combination of PAMs (*Figure 4B*). The synergistic action of BINA and VU0415374 was also observed in a functional cAMP assay (*Figure 4D*) and constitutes therefore a third fingerprint for the mGlu2-4 heterodimers.

## mGlu2 and mGlu4 can form heterodimers in neurons

In primary cultures of hippocampal neurons, co-expression of CLIP-mGlu2 and SNAP-mGlu4 subunits could be detected at the cell surface through labeling with CLIP and SNAP substrates carrying either Lumi4-Tb or Red. In these neurons, using a lanthanide-based time-resolved FRET microscope that we recently developed (*Faklaris et al., 2015*), we detected a TR-FRET signal between the CLIP and SNAP subunits equivalent to that measured for homodimeric mGlu2 receptors. This observation is consistent with the formation of mGlu2-4 heterodimers in transfected neurons (*Figure 5*). In contrast, no TR-FRET could be detected between CLIP-mGlu2 and SNAP-mGlu1 (*Figure 5*), two subunits reported not to associate into heterodimeric entities (*Doumazane et al., 2011*; *Levitz et al., 2016*). Such mGlu2-4 heterodimers are not the consequence of a large over-expression of these tagged subunits, since their quantification using a mGlu2 specific antibody, relative to the endogenous mGlu2 revealed a five times higher expression of the tagged receptor only (Möller et al., manuscript submitted for publication).

## Functional mGlu2-4 heterodimers in a neuronal cell line

As a first attempt to identify native mGlu2-4 heterodimers in non-transfected cells, we used the striatal cell line STHdh[Q7] (*Trettel et al., 2000*), since both mGlu2 and mGlu4 mRNA have been reported in the striatum, although to a low level (*Ohishi et al., 1993*; *Conn et al., 2005*; *Ferraguti and Shigemoto, 2006*; *Gu et al., 2008*; *Beurrier et al., 2009*). We first examined whether mGlu2-4 heterodimers can form in these cells upon transfection of CLIP-mGlu2 and SNAP-mGlu4. The formation of mGlu2-4 heterodimers was well illustrated by the cooperativity between the binding sites characteristic of the heterodimer, as revealed with the TR-FRET sensor assay (*Figure 6—figure supplement 1*). We then examined whether a native mGluR-mediated responses with the characteristics of the mGlu2-4 heterodimer could be detected in these cells. Unfortunately, the endogenous receptor levels are very low and the detection of their activation could not be detected using cAMP assays. However, we achieved measuring responses with mGlu2 and mGlu4 agonists using the xCELLigence technique, a label-free method reporting on small variations in cell shape (cell index). STHdh cells are shrunk when adenylate cyclase is activated (*Figure 6A*). Taking advantage of this characteristic we observed that mGlu2- and mGlu4-specific ligands impaired the forskolin effect in a pertussis toxin-sensitive way, consistent with the presence of endogenous Gi-coupled mGlu2 and mGlu4 receptors in these cells (*Figure 6—figure supplement 2*). Both LY354740 and the mGlu4 preferential agonist LSP4-2022 (*Goudet et al., 2012*) produced a dose-dependent effect with the expected $EC_{50}$s (*Figure 6B*).

As described above with the mGlu2-4 heterodimer (*Figure 3*), LSP4-2022 increased the potency of the mGlu2 agonist LY354740 (*Figure 6B,C*). This mGlu4 effect is clearly due to the presence of mGlu4 in these STHdh cells, since depletion of mGlu4 with a lentivirus expressing mGlu4 ShRNA,

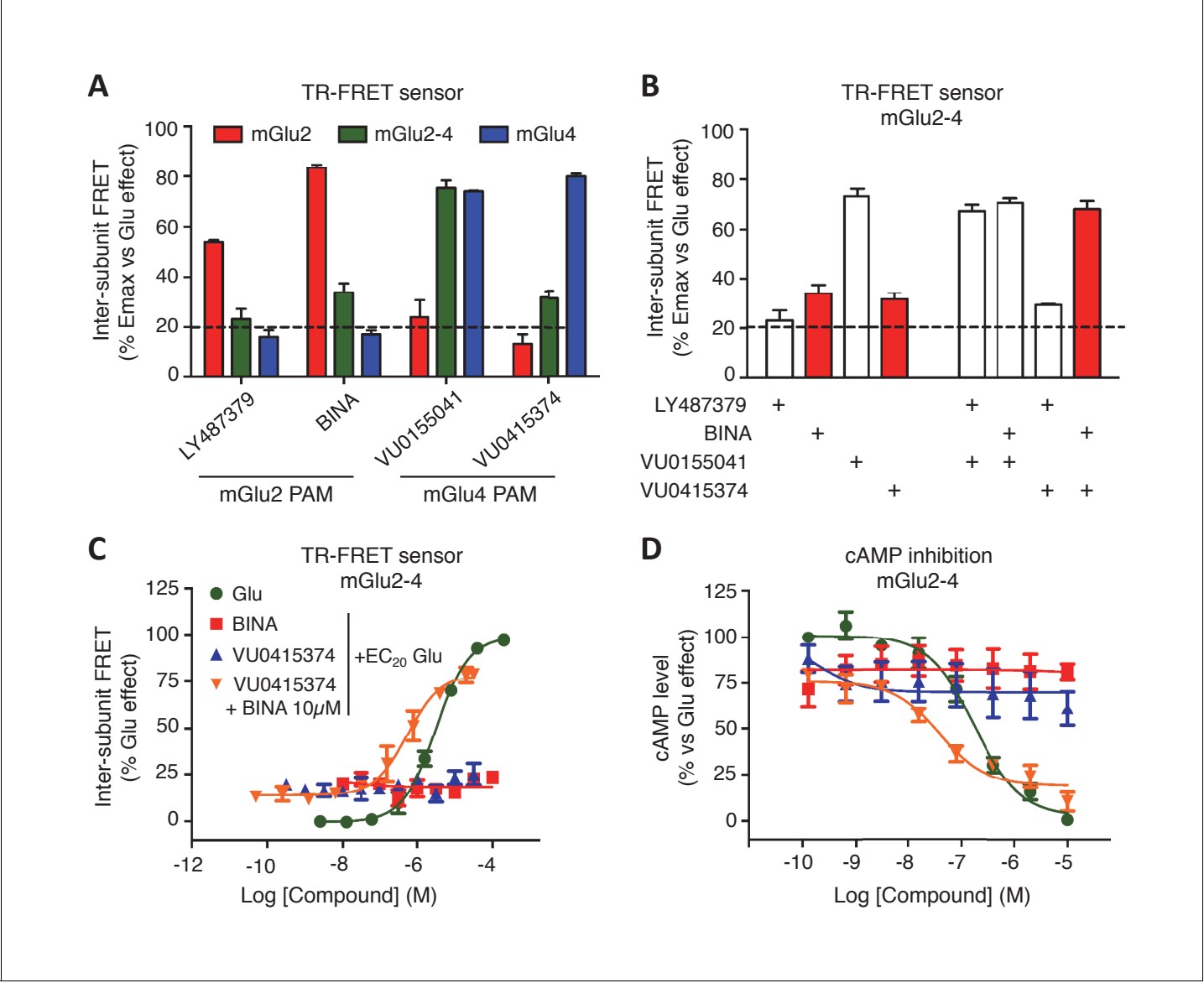

**Figure 4.** Synergistic action of mGlu2 and mGlu4 PAMs in mGlu2-4 heterodimer. (A) Effect of mGlu2 (LY487379 10 µM, BINA 10 µM) and mGlu4 (VU0155041 10 µM, VU0415374 10 µM) PAMs on mGlu2 (red), mGlu4 (blue) and mGlu2-4 (green) TR-FRET sensors in the presence of an $EC_{20}$ of glutamate. (B) Effect of mGlu2 and/or mGlu4 PAMs on the response mediated by an $EC_{20}$ concentration of glutamate in mGlu2-4 heterodimer by TR-FRET. The strong synergy between BINA and VU0415374 is highlighted with red bars. (C–D) Dose response of BINA and/or VU0415374 in potentiating the effect of $EC_{20}$ glutamate on TR-FRET sensors (C) and cAMP inhibition (D). Results are mean ± SEM of three independent experiments performed in triplicates.

resulted in a higher potency of LY354740 consistent with its potency on mGlu2 homodimers, with no further effect of the mGlu4 agonist (*Figure 6C*). Furthermore, the synergistic effect of BINA and VU0415374 observed on the mGlu2-4 heterodimer (*Figure 4*) could also be observed in the STHdh cells (*Figure 6D*). Indeed, either BINA (1 µM) or VU0415374 (1 µM) modestly potentiated the effect of agonists (a combination of 10 nM LY354740 and 100 nM LSP4-2022), while a strong potentiation was observed with these two PAMs added simultaneously (*Figure 6D*). No such effect was observed in the mGlu4-silenced cells (*Figure 6D*). These data are consistent with the existence of mGlu2-4 heterodimers endogenously expressed in the STHdh cells.

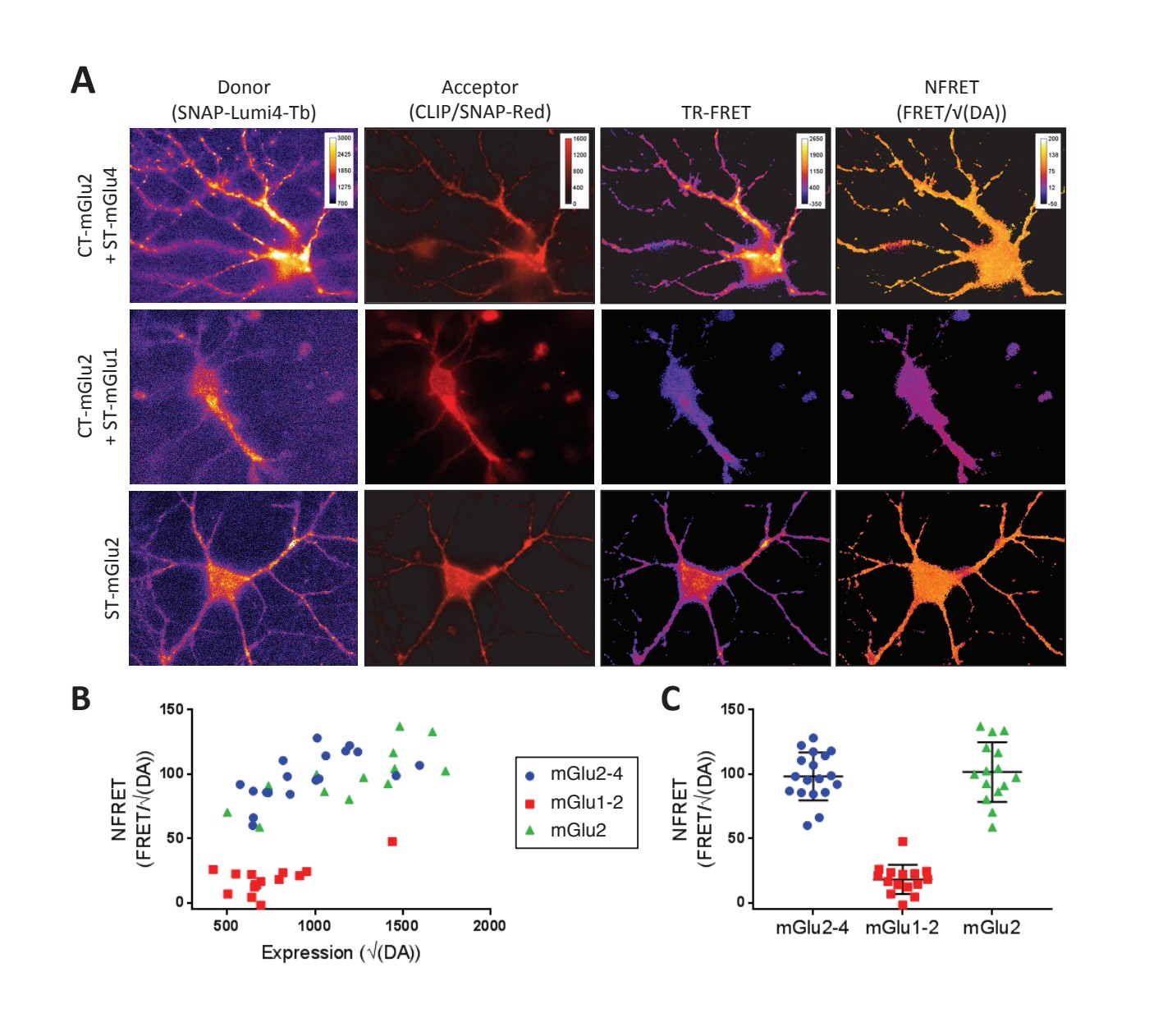

**Figure 5.** TR-FRET detection of mGlu2-4 heterodimers in transfected hippocampal neurons. Neurons transfected with CLIP-mGlu2 and SNAP-mGlu4 are compared with either CLIP-mGlu2 and SNAP-mGlu1 (negative control) or SNAP-mGlu2 (positive control). The receptors are labeled with Lumi4-Tb as donor and Red as acceptor. (**A**) Image examples of neurons expressing the three receptor combinations at comparable expression levels showing similar TR-FRET and NFRET (TR-FRET normalized to the expression of donor and acceptor) signals for the mGlu2-4 heterodimer and the mGlu2 homodimer and a low signal for the mGlu1-2 heterodimer. TR-FRET images are corrected for bleedthrough and thresholded to remove background and noise. Images are 138 μm wide. (**B**) Quantification of NFRET as a function of the expression level of donor and acceptor. Each point is the quantification of one neuron. (**C**) Scatter plot of NFRET.

## Pharmacological evidence for mGlu2-4 receptors in lateral perforant path terminals

In the hippocampus, mGlu4 receptor expression is prominent in the inner third of the molecular layer of the dentate gyrus (*Shigemoto et al., 1997*; *Corti et al., 2002*). mGlu2 receptor is also expressed in the molecular layer of the dentate gyrus (*Ohishi et al., 1993*; *Gu et al., 2008*; *Wright et al.,*

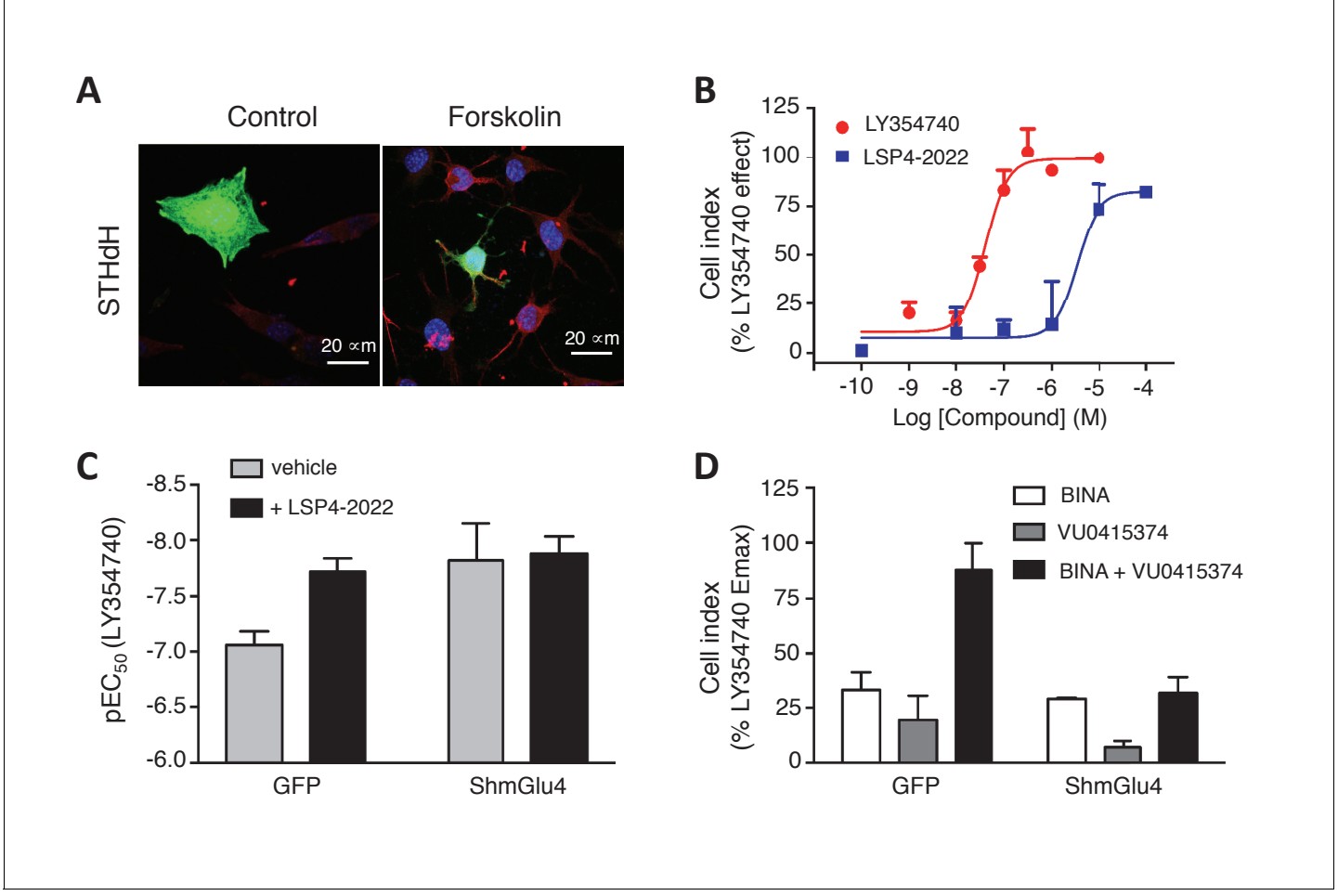

**Figure 6.** Functional evidence for mGlu2-4 heterodimers in a neuronal cell line, STHdh. (**A**) Representative image of neuronal cells treated or untreated with forskolin. Cells were transfected with GFP (green) and stained with MAP2 (red) and DAPI (blue). (**B**) Dose response on LY354740 or LSP4-2022 represented in % variation of cell index versus LY354740. (**C**) pEC50 values of LY354740 as determined using the change in cell index, in control cells transfected with GFP, and in cells transfected with the SH RNA against mGlu4, under control condition (gray bars) or in the presence of the mGlu4 agonists LSP4-2022 (10 μM). (**D**) Maximum effect of mGlu2 (BINA 1 μM) and/or mGlu4 PAM (VU0415374 1 μM) in potentiating the effect of low concentration of LY354740 (10 nM) and LSP4-2022 (100 nM) in control STHdh cells infected with GFP vector or silencing shRNA for mGlu4. Data in **B–D** are means ± SEM of three independent experiments performed in triplicates.

The following figure supplements are available for figure 6:

**Figure supplement 1.** mGlu2-4 heterodimer TR-FRET sensor transfected in SThdH striatal cell line.

**Figure supplement 2.** mGlu2 and mGlu4 ligands impair forskolin shrinking of SThdH striatal cell line.

*2013*). As expected, activation of mGlu2 (concentration >300 nM LY354740) or mGlu4 receptor (concentration >5 μM LSP4-2022) inhibited synaptic transmission at the LPP (*Figure 7*). The effect of LSP4-2022 is absent in slices prepared form mGlu4 KO mice, demonstrating its effect is mediated by mGlu4 in control animals (*Figure 7*)

In order to investigate whether mGlu2-4 heterodimers are expressed in these synapses, we applied a low concentration of each agonist that produced no detectable inhibitory effect in the LPP. Only when co-applied, LSP4-2022 (100 nM, 3.73 ± 2.19%, n = 4) and LY354740 (10 nM, 2.62 ± 3.83%, n = 5) induced a significantly large reduction of the fEPSPs in LPP (20.05 ± 5.82%; n = 6; *Figure 7C*). No such effect was observed in slices prepared from mGlu4 KO mice (*Pekhletski et al., 1996*) (*Figure 7*). The synergy of these agonists is consistent with the presence of mGlu2-4

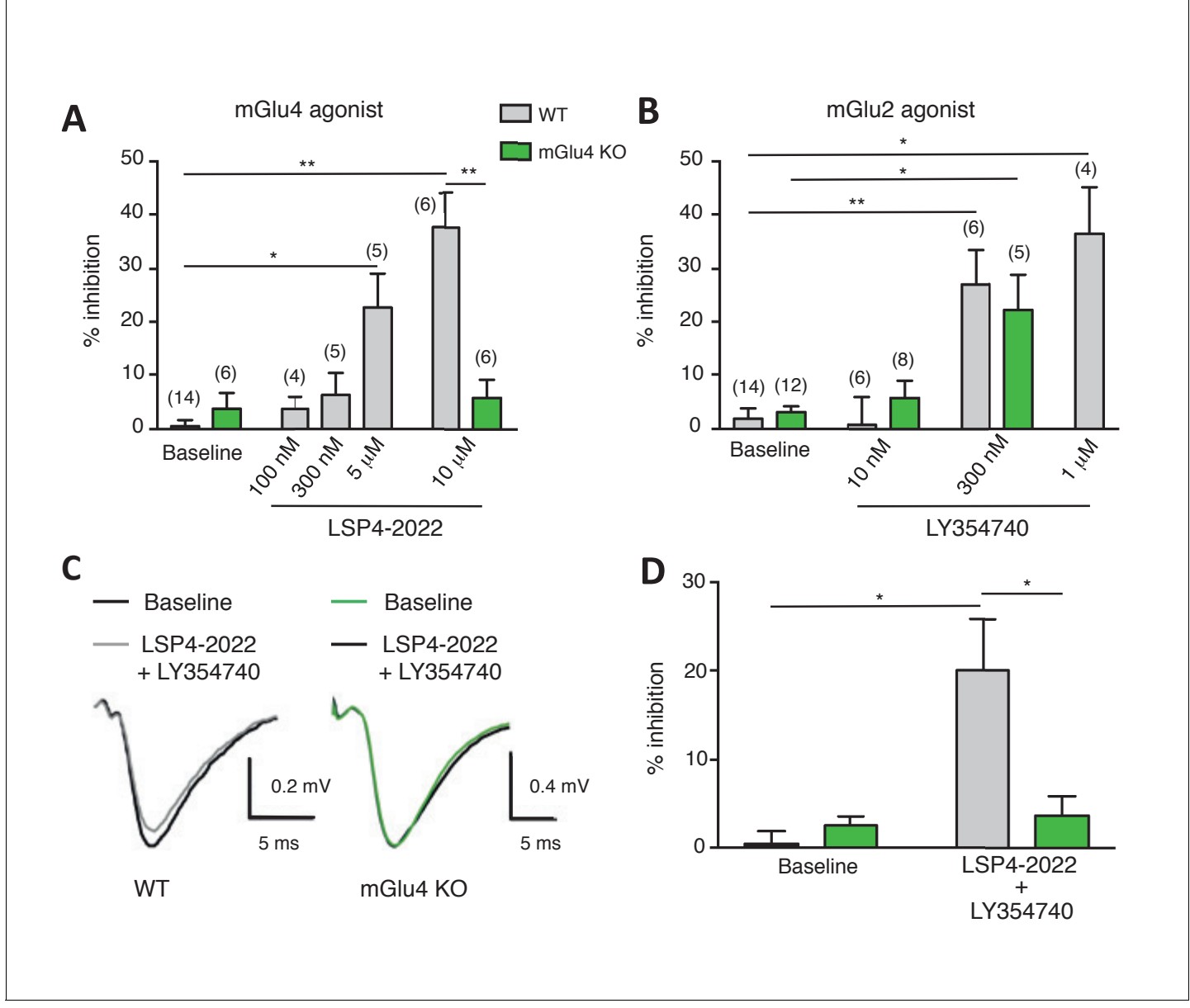

**Figure 7.** Effect of LY354740 and LSP4-2022 in the LPP of wild-type (WT) mice. (**A**) Bar graph illustrating the % inhibition of fEPSPs induced by low (100 nM and 300 nM) and high (5 μM and 10 μM) concentrations of LSP4-2022 in the LPP. Only high concentrations of LSP4-2022 induced a significant decrease of fEPSP amplitude. Green bars indicate data obtained using slices from mGlu4 KO mice. (**B**) Inhibitory effect of LY354740 on fEPSP amplitude in the LPP. Note that 300 nM or 1 μM LY354740 caused a significant decrease of fEPSP amplitude. Green bars indicate data obtained using slices from mGlu4 KO mice. (**C**) Representative averaged traces of evoked synaptic activity induced by LPP stimulation in field recording of granular cells from WT mice (*Left*). Bar graph illustrating the % inhibition of fEPSP amplitude by LY354740 (10 nM), LSP4-2022 (100 nM) and LY354740 (10 nM) / LSP4-2022 (100 nM) in the LPP (*Right*). Note that application of LY354740 (10 nM) + LSP4-2022 (100 nM) significantly decreased the fEPSP amplitude. Data in A-C are means ± SEM of (n) independent experiments from at least three different animals. *p<0.05, **p<0.001.

The following figure supplement is available for figure 7:

**Figure supplement 1.** Absence of synergistic effect between mGlu2 and the Gi-coupled delta opioid receptor.

heterodimers in these terminals. However, one cannot exclude a possible synergy between mGlu2 and mGlu4 at the signaling level, rather than within a heterodimer. We think this is unlikely since such strong synergistic effect have not been observed between Gi-coupled receptors. Indeed, upon co-expression of mGlu2 and the Gi-coupled delta opioid receptor (DOR), activation of DOR with SNC162 had no effect on the potency of LY354740 in inhibiting cAMP formation via mGlu2 receptors (*Figure 7—figure supplement 1*).

## Discussion

Despite their description in heterologous cells 5 years ago (*Doumazane et al., 2011*), evidence for the existence of mGlu heterodimers in vivo remains elusive. Using two different approaches to characterize mGlu2-4 heterodimers specifically, we identified pharmacological fingerprints of such receptors. First, the mGlu2 selective agonist LY354740 behaves differently on the mGlu2-4 heterodimer than on the mGlu2 homodimer, including a lower potency, and a lower Hill coefficient. Such complex properties of LY354740 disappeared in the presence of mGlu4 agonists. Second, mGlu4 agonists largely increase the potency of LY354740. Third, among four mGlu2 or mGlu4 PAMs tested, only VU0155041 potentiated the effect of agonists on the mGlu2-4 heterodimer. Fourth, a combination of two PAMs (BINA and VU0155041) inactive when applied alone enhanced agonist action on the heterodimer when applied together. Such pharmacological fingerprints provide ways for demonstrating the existence of such heterodimers in native cells, as illustrated here with a neuronal cell line and the medial perforant path terminals in the dentate gyrus.

One major difficulty in studying the functional and pharmacological properties of GPCR heterodimers is the ability of each subunit to form functional receptors, making difficult the measurement of signals originating from the heterodimers exclusively. In previous studies, properties of the mGlu2-4 heterodimers were studied in cells co-expressing both mGlu2 and mGlu4 (*Yin et al., 2014*). However, as illustrated in *Figure 1—figure supplement 1*, even when conditions were used for an optimal expression of the heterodimer at the cell surface, data obtained were always contaminated by responses mediated by the homodimers. In this study, we developed two different approaches that allowed the analysis of the pharmacological and functional properties of mGlu2-4 heterodimers, both strategies being likely useful for other class C heterodimers. Of note, the TR-FRET sensor assay that relies on the inter-subunit movement in mGlu dimers allows the specific analysis of compounds in any of the three combinations specifically. This allowed us to identify specific properties of the heterodimers that can be useful for the characterization of such receptors in native cells.

Our data show that agonist occupancy of both subunits is required for a full activity of the heterodimer, as well illustrated using specific agonists of one subunit (either mGlu2 or mGlu4), or by mutating the binding site of either subunit. Accordingly, activating either the mGlu2 or mGlu4 VFT in the mGlu2-4 heterodimer leads to a similar partial effect, both at the conformational level of the VFT dimer, as revealed with the TR-FRET sensor assay, and at the signaling level. This finding is consistent with previous studies demonstrating that two agonists are required, with both VFT closed to fully activate mGlu5 homodimers (*Kniazeff et al., 2004*). However, it was reported that the low activity observed when a single protomer is occupied by an agonist in mGlu2 homodimers possibly results from the spontaneous closure of the second, unliganded VFT (*Levitz et al., 2016*). It is therefore possible that part of the activity observed with mGlu2 selective agonists on mGlu2-4 is due to a spontaneous closure of the mGlu4 VFT. This is consistent with the biphasic curve of DCG-IV, an mGlu2 agonist that has mGlu4 antagonist activity at high concentration.

The effect of LY354740, a well-known group-II-specific mGluR agonist, appears quite complex on the mGlu2-4 heterodimer. This compound has a partial efficacy, but a low Hill coefficient on the heterodimer. Its mGlu2 potency and normal Hill coefficient are restored when the mGlu4 subunit is either activated or mutated to prevent ligand binding. These findings cannot be explained by a simple model in which both mGlu2 and mGlu4 ligands bind selectively to their respective subunit. Instead, our data suggest that LY354740 can bind to both subunits, its interaction with mGlu2 increasing its affinity to mGlu4 in the heterodimer. Such ligand cooperativity between bindings sites in an mGlu dimer has already been reported in mGlu5 receptors in which one binding site is mutated (*Kniazeff et al., 2004*; *Rovira et al., 2008*). The synergistic effect observed between mGlu2 and mGlu4 agonists appears as an interesting property to identify mGlu2-4 heterodimers in neurons.

When examining the effect of two mGlu2 (BINA and LY487379) and two mGlu4 PAMs (VU0155041 and VU0415374), we confirmed that VU0155041 is the only PAM able to potentiate the effect of glutamate on the mGlu2-4 heterodimer (*Yin et al., 2014*). Although BINA and VU0415374 had very modest effects alone, their co-application largely potentiated the heterodimer. The structural basis for this synergistic effect, especially when considering that a single 7TM is active at a time in mGlu dimers (*Hlavackova et al., 2005*, *2012*), remains unclear and will be the subject of further studies. Whatever the reason, this synergistic effect between these two PAMs offers another way to identify mGlu2-4 heterodimers in neurons.

A recent study revealed that mGlu2-4 heterodimers are likely present in cortico striatal terminals (*Yin et al., 2014*). This conclusion is based on co-immunoprecipitation data, and on the lack of effect of PHCCC, an mGlu4 PAM devoid of effects in cells co-expressing mGlu2 and mGlu4, while VU0155041 that is active on both mGlu4 and mGlu2-4 heterodimers, potentiated the response (*Yin et al., 2014*). Our data also revealed that mGlu2-4 can form in transfected neurons, indicating there are no specific mechanisms in neurons that would prevent the formation of such heterodimers. Most importantly, we found that in a neuronal cell line, responses with the pharmacological characteristics of the mGlu2-4 heterodimers can be recorded. Indeed, the synergistic effects of mGlu2 and mGlu4 ligands (both agonists and PAMs) typical of the mGlu2-4 heterodimer were observed in these cells, bringing strong evidence that endogenous mGlu2-4 heterodimers exist in these neuronal cells despite the low expression of both mGlu2 and mGlu4. The synergistic activity of the agonists LY354740 and LSP4-2022 was also observed in the terminals of the medial perforant path in the dentate gyrus where both mGlu2 and mGlu4 subunits are present (*Shigemoto et al., 1997*). Such a synergistic activity is no longer observed in slices prepared from mGlu4 KO mice, demonstrating the involvement of mGlu4. However, we cannot rule out that such a synergy may come from the signaling cascades activated by both mGlu2 and mGlu4 homodimers. We still think this is unlikely because such a strong synergistic effect has not been observed between Gi-coupled receptors and indeed could not be observed between mGlu2 and the delta opioid receptor co-expressed in the same cells.

It is sometimes argued that GPCR dimers and heterodimers result from the overexpression of the partners. For several reasons, this is unlikely the case for the mGlu2-4 heterodimer. Over-expression is expected to result in larger mGluR complexes since mGluRs are constitutive and covalent dimers (*Calebiro et al., 2013*), and no proximity could be detected by FRET between mGlu2 and mGlu1 receptors known not to form heterodimers (*Doumazane et al., 2011*), even in transfected neurons. The FRET efficacy between mGlu2 and mGlu4 largely decrease upon receptor activation, as expected for a correctly assembled dimer. The relative quantification of transfected over endogenous receptors revealed a fivefold only over-expression (Møller et al., manuscript in preparation). Various approaches have been used to estimate the size of the mGlu2-4 complex and all revealed a strict dimer (*Doumazane et al., 2011*). Eventually, a receptor with the pharmacological fingerprints of the mGlu2-4 heretodimer was observed in STHdh cells where both mGlu2- and mGlu4-mediated responses were difficult to detect, suggesting a low expression level.

Taken together, our data add to previous studies suggesting the existence of mGlu heterodimers in the brain. We show that mGlu2-4 receptors likely exist in the brain and we report innovative approaches that will be useful to confirm the existence of other mGlu heterodimers. For example, one may propose the existence of heterodimers containing an mGlu7 subunit, for which the very low glutamate potency raised a number of questions regarding its roles in vivo. Indeed, mGlu7 can be found with other high-affinity mGluRs, such as mGlu8 (*Ferraguti et al., 2005*), offering a way to involve mGlu7 in a receptor heterodimer with specific properties. Proteomic experiments also identified mGlu5 as a partner of mGlu1 (*Pandya et al., 2016*), a finding that likely explains surprising functional studies regarding the effect of specific mGlu1 and mGlu5 inhibitors in the hippocampus (*Huber et al., 2001*; *Volk et al., 2006*). Functional studies also suggested mGlu3 receptors could be involved in heterodimeric entities with mGlu2 receptors (*Iacovelli et al., 2009*). A clear view of such mGlu heterodimers is definitively needed since all possible combinations observed in transfected cells suggest the existence of 16 additional receptor entities in the brain. In addition, first results are already highlighting specific roles played by homo and heterodimers in the actions of drugs with therapeutic potentials. For example, PAMs selective for the homodimeric mGlu4 may be preferred for the treatment of Parkinson disease (*Niswender et al., 2016*). Our study highlighting techniques

to decipher the specific properties of mGlu heterodimers will definitively help solving these important issues.

## Materials and methods

### Materials

SNAP-Lumi4-Tb, SNAP-Green and CLIP-Green were from Cisbio Bioassays (Codolet, France). SNAP-block and CLIP-block were from New England Biolabs (Ipswich, MA, USA). (2R,4R)-APDC, ACPT-I, L-AP4, BINA, DCG-IV, LY341495, LY354740, LY487379, VU0155041 and SNC162 were purchased from Tocris Bioscience (Bristol, UK). LSP4-2022 was kindly provided by Dr. Francine Acher (Université Paris Descartes, France). VU0415374 was synthesized by Dr. Xavier Gómez and provided by Dr. Amadeu Llebaria (University of Barcelona, Spain). Control GFP and ShmRNA mGlu2 and mGlu4 lentiviral particles were purchased from Santa Cruz Biotechnology Inc. (Dallas, TX, USA). The mGlu4 KO mice were obtained from Dr David Hampson (Toronto, Canada [*Pekhletski et al., 1996*]), and their genotype determined as reported (*Pitsch et al., 2007*).

### Plasmids

The pRK5 plasmids encoding the wild-type rat mGlu subunits in which the SNAP or CLIP has been inserted at their N-term after the signal peptide, and constructs with YADA mutations in mGlu2 were previously described (*Doumazane et al., 2011*, *2013*). The pRK5 plasmid encoding for the ligand binding deficient SNAP-mGlu4-YADA mutant in which the two residues Y230 and D312 important for agonist binding in the VFT were mutated, was generated by site-directed mutagenesis using QuikChange mutagenesis protocol (Agilent Technologies, Les Ulis, France) using the SNAP-mGlu4 plasmid as a template (*Doumazane et al., 2011*). The sequence coding C1 (the 47-residue coiled-coil sequence of the C-terminal of GABA$_{B1}$), or C2 (the 49-residue coiled-coil region of GABA$_{B2}$), followed by the endoplasmic reticulum retention signal KKTN, as previously described (*Huang et al., 2011*), was used to generate the constructs SNAP-mGlu4-C1 and CLIP-mGlu2-C2. SNAP-mGlu4-C1 was obtained by replacing the last 38 residues in mGlu4 C-term (SNAP-tagged version of mGlu4 was used) by C1KKTN. In this construct, the C-term of SNAP-mGlu4-C1 is . . . NKFT*TG SSTNNNEEEKSRLLEKENRELEKIIAEKEERVSELRHLQSRQQLKKTN* (the last residues (up to Thr874) of mGlu4 are underlined, those of C1 are in italic). The C-term sequence of CLIP-mGlu2-C2 was previously described (*Huang et al., 2011*). The plasmid encoding SNAP-delta opioid receptor was from Cisbio Bioassays.

### Cell culture and transfection

HEK293 cells (ATCC, CRL-1573, lot: 3449904) were cultured in Dulbecco's modified Eagle's medium (Thermo Fischer Scientific, Courtaboeuf, France) supplemented with 10% (vol/vol) fetal bovine serum (Sigma Aldrich) in a P2 cell culture room. Absence of mycoplasma was routinely checkedusing the MycoAlert Mycoplasma detection kit (LT07-318 (Lonza, Amboise, France), according to the manufacturer protocol. HEK 293 cells were used after 35 to 40 passages and transfected with a reverse transfection protocol using Lipofectamine 2000 (Thermo Fischer Scientific, Courtaboeuf, France), and finally plated in polyornithine-coated, black-walled, dark-bottom, 96-well plates at $10^5$ cells/well. To avoid too high concentrations of glutamate in the assay medium that could interfere with mGluR activity, cells were cotransfected with the plasmid encoding the glutamate transporter EAAC1 and incubated in DMEM Glutamax medium (Thermo Fischer Scientific) at least 2 hr before the different assays were performed. Frozen-labeled HEK-293 cells were transfected as described above, labeled as described below, then frozen at −80°C with 10% DMSO and fetal bovine serum, and later washed three times in Krebs buffer (10 mM Hepes pH 7.4, 146 mM NaCl, 4.2 mM KCl, 1 mM CaCl$_2$, 0.5 mM MgCl$_2$, 5.6 mM glucose, bovine serum albumin (BSA) 0.1%) before use.

In order to optimize the best expression of mGlu2-mGlu4 heteromers, several ratios of mGlu2: mGlu4 were assayed. It was determined by TR-FRET analysis that 2:1 ratio (40 ng CLIP-mGlu2: 20 ng SNAP-mGlu4) was optimal for the detection of all populations (*Figure 1—figure supplement 1A–C*). Using these conditions, a large batch of cells were transfected, labeled and frozen to perform a complete screening of the different compounds in 384-well plates.

Conditionally immortalized wild-type STHdh$^{Q7}$ striatal neuronal progenitor cell line (*Trettel et al., 2000*) were kindly provided by Dr Sílvia Ginés (University of Barcelona, Spain). These cells nicely differentiated and became MAP2 positive when cultured in a differentiated medium as described (*Trettel et al., 2000*). We also verified that they were still responsive to dopamine D1 and histamine H3 receptor agonists using the Xcellingence technology. Neuronal cells were grown at 33°C in DMEM (Sigma-Aldrich), supplemented with 10% fetal bovine serum (FBS), 1% streptomycin-penicillin, 2 mM L-glutamine, 1 mM sodium pyruvate, and 400 µg/ml G418 (Geneticin; Invitrogen). Neuronal cells were transfected with Lipofectamine LTX (Thermo Fischer Scientific, Courtaboeuf, France) following the protocol from the provider. To perform silencing of mGlu2 and mGlu4, STHdh cells were infected with control GFP vector, ShmRNA mGlu2 or mGlu4 vector and after 48 hr, infected cells were selected by adding hygromycin-containing medium.

## Fluorescence labeling and TR-FRET measurements

SNAP-tag labeling alone and orthogonal labeling of SNAP- and CLIP-tag were performed as described previously (*Scholler et al., 2017*). Briefly, for SNAP-tag labeling, 24 hr after transfection, HEK293 cells were incubated at 37°C for 1 hr with a solution of 100 nM of SNAP-Lumi4-Tb, 60 nM of SNAP-Green and 1 µM CLIP-block, in case of FRET detection between SNAP-tag subunits. For CLIP labeling, cells were incubated with 1 µM CLIP-Lumi4-Tb, 800 nM CLIP-Green and 1 µM SNAP-block. For co-labeling of the SNAP- and CLIP tags, cells were incubated at 37°C for 2 hr with a solution of 300 nM SNAP-Lumi4-Tb and 1 µM CLIP-Green. After labeling, cells were washed three times with Krebs buffer, and drugs were added. Then, the TR-FRET measurements were performed on a PHER-Astar FS microplate reader (BMG Labtech, Ortenberg, Germany) which is standardly equipped with 'TR-FRET' optical modules and two photomultiplier tubes to detect two emission wavelengths representing donor and acceptor emission simultaneously, as previously described (*Scholler et al., 2017*). To monitor the emissive decay curves, the Lumi4-Tb present in each well was excited using N$_2$ laser emission line at 337 nm (40 flashes per well for the 96-well plate format, 20 flashes per well for the 384-well plate format). The emission decay was collected during 2500 or 5000 µs with 5 µs or 10 µs steps, respectively, at 620 nm for the donor (Lumi4-Tb) and at 520 nm for Green, as can be indicated in the 'advanced mode' option of the plate-reader's software. For acceptor ratio determination, optimal integration windows were determined as previously reported (*Scholler et al., 2017*). The acceptor ratio was calculated using the sensitized acceptor signal integrated over the time window [50 µs-100 µs], divided by the sensitized acceptor signal integrated over the time window [800 µs-1200 µs].

## cAMP functional assay

The amount of cAMP was determined using the Glosensor$^{TM}$ cAMP assay (Promega Corporation, Madison, USA). HEK293 cells were co-transfected with the indicated mGluR plasmids and the pGlo-Sensor-22F plasmid. The day after, cells were starved during 2 hr in serum-free medium and afterwards incubated in Krebs buffer with 450 µg/ml luciferin (Sigma-Aldrich) during 1 hr. The luminescence peak signal was measured on a Mithras microplate reader at 28°C during 8 min since luminescence signal was stable. Then, forskolin plus mGluR ligands were added and luminescence was measured for 30 min.

## Label-free impedance assay

xCELLigence plates were coated with poly-ornithine and laminin during 1 hr, and neuronal cells were seeded at $3 \times 10^4$ cells/well and introduced into the incubator at 33°C overnight. Medium was replaced by serum-free medium during 2 hr to reach a stable cell index, and then forskolin and mGluR ligands were added and the signal was followed during at least 2 hr using xCELLigence RTCA DP apparatus (ACEA Bioscience Inc, San Diego, USA). When antagonists or PAMs were used, they were added 20 min before forskolin. Pertussin toxin was added 4 hr after cell plating and incubated overnight.

## Neuronal culture and TR-FRET microscopy

Hippocampi from Sprague-Dawley rat embryos on embryonic day 18 (E18) were dissected, dissociated by treatment with liberase TL (Roche, Boulogne-Billancourt, France), then mechanical triturated and plated on Lab-Tek II chambered cover slides (Thermo Fisher Scientific) coated with poly-L-

ornithine and laminin (Sigma-Aldrich) at a density of ~300 neurons/mm$^2$. Neurons were cultured in Neurobasal medium (Thermo Fisher Scientific) supplemented with 2% B-27 (Thermo Fisher Scientific), 100 U/ml Penicillin-Streptomycin (Thermo Fisher Scientific), 10 mM HEPES, and 0.5 mM Glutamax medium (Thermo Fisher Scientific). 0.5 mM L-glutamine was added when plating the cells. Half of the medium was exchanged weekly. Neurons were transfected with Lipofectamine 2000 at 10 days in vitro (DIV). The medium was exchanged after 4 hr of incubation with the transfection reagent with half fresh medium and half medium conditioned by incubation with primary neurons. pRK5 plasmids for expression of SNAP- or CLIP-tagged rat mGluRs under control of the CMV promoter were previously described (*Doumazane et al., 2011*). For increased expression, the CMV promoter was exchanged with the synapsin-1 promoter (gift from B. Bettler) for CLIP-mGlu2 and SNAP-mGlu4. Homo- and heterodimers were expressed by co-transfection with CLIP-mGlu2 (100 ng/well) + SNAP-mGlu4 (200 ng/well), CLIP-mGlu2 (200 ng/well) + SNAP-mGlu1a (100 ng/well), or SNAP-mGlu2 (300 ng/well). For TR-FRET microscopy, 16–17 DIV neurons were labeled with 100 nM SNAP-Lumi4-Tb + 1000 nM CLIP-Red (heterodimers) or 100 nM SNAP-Lumi4-Tb + 500 nM SNAP-Red for 1 hr at 37°C in imaging buffer (10 mM Hepes pH 7.4, 127 mM NaCl, 2.8 mM KCl, 1.1 mM MgCl$_2$, 1.15 mM CaCl$_2$, 10 mM glucose) supplemented with 1% BSA followed by a wash in imaging buffer with 1% BSA and three washes in imaging buffer. Cells were imaged in imaging buffer. Images were acquired with a homebuilt TR-FRET microscope (*Faklaris et al., 2015*). Briefly, the donor was excited with a 349 nm Nd:YLF pulsed laser at 300 Hz with ~68 µJ/pulse followed by collection of either the donor signal using a 550/32 nm bandpass filter or the TR-FRET signal using a 700/75 nm bandpass filter. In both cases, images were acquired with 10 µs delay between excitation and collection of emission, 3 ms acquisition time and 4000 acquisitions. The acceptor was excited with a mercury lamp using a 620/60 nm bandpass filter, and the emission was collected for 300 ms with a 700/75 nm bandpass filter. Time-gated images were shading corrected by dividing the raw image with a background image using ImageJ version 1.51 f (*Schneider et al., 2012*). Correction for donor bleedthrough (6%) and generation of NFRET images (TR-FRET/(donor × acceptor)$^{0.5}$) was done with the PixFRET plugin to ImageJ (*Feige et al., 2005*). Acceptor bleedthrough and direct acceptor excitation was not detected. For quantification, all non-zero pixels in the NFRET image were selected, pixels not belonging to the cell removed and the modal NFRET value and mean donor signal were measured for this selection.

## Electrophysiological recordings

Acute slices were prepared from adult (P21-P30) control or mGlu4-KO mice following a protocol approved by the European Communities Council Directive and the French low for care and use of experimental animals. Mice were decapitated, and brains quickly removed and chilled in cold artificial cerebro-spinal fluid (ACSF) containing 125 mM NaCl, 2.5 mM KCl, 1.25 mM NaH$_2$PO$_4$, 25 mM NaHCO$_3$, 1 mM MgCl$_2$, 2 mM CaCl$_2$, and 10 mM glucose, pH 7.4, equilibrated with 95% O$_2$ and 5% CO$_2$. Parasagittal acute 400-µm-thick slices were prepared with a vibratome (7000 smz, Campden Instruments LTD, England) in ice-cold ACSF. Sections were kept at room temperature for at least 1 hr before recording. Slices were transferred to a submersion recording chamber, maintained at 30°C and perfused with oxygenated ACSF at a rate of one chamber volume (1.5 ml) per minute.

fEPSPs were evoked at 0.033 Hz using bipolar stimulating electrode and recorded using glass micropipettes (3–5 MΩ) and filled with 3 M NaCl. Stimulating electrode was placed in the outer thirds of the molecular layer of the dentate gyrus for stimulation of the lateral perforant path (LPP). Correct positioning of electrodes was verified by application of paired-pulse at an interval of 100 ms induces paired-pulse facilitation in the LPP. The effect of paired-pulse stimulation was assessed and only those slices that displayed the correct facilitation in the LPP were used for this study. Input-output curves were generated for each slice, and the stimulation intensity was adjusted to 70% of the maximum response. Baseline fEPSPs were recorded for a minimum of 20 min before bath-application of different agonists or PAMs. Evoked responses were analyzed by measuring the slope of individual fEPSPs. The slopes from two sequential sweeps were averaged. All slopes were normalized to the average slope calculated during the pre-drug period (percentage of baseline). All data were analyzed offline using pClamp 9 (Molecular Devices) and are reported as the mean ± SEM. Statistical comparisons were made using two-tailed unpaired or paired Student's t-tests. Differences were considered significant at $p < 0.05$. The % inhibition was calculated by the difference of the slope between the baseline and the last 4 min of the drug application.

## Curve fitting and data analysis

SAS/STAT 9.4 (SAS Institute, Cary, NC, USA) statistical package was used for parameter optimization and statistical analyses in mathematical modeling. Curve fitting was performed using nonlinear regression using GraphPad Prism software. p-values<0.05 were considered statistically significant using one-way ANOVA with Bonferroni post-hoc test.

## Acknowledgements

This work was made possible thanks to the Arpege facility from BioCampus (IGF, Montpellier, France). The authors wish to thank Drs A Llebaria and X Gomez (Barcelona, Spain), and F Acher (Paris, France) for providing VU0415374 and LSP4-2022, respectively; Dr Sílvia Ginés (University of Barcelona, Spain) for providing the STHdh[Q7] cells; and Dr Bernhard Bettler (Pharmacentrum, Basel, Switzerland) for the synapsin-1 promoter. The authors also thank Michaël Mathieu for the delta opioid receptor experiments. This work was supported by the Centre National de la Recherche Scientifique, the Institut National de la Recherche Médicale, the University of Montpellier, Cisbio Bioassays and by the Fondation pour la Recherche Médicale (grant no. DEQ20130326522) to JPP, and by the Agence Nationale de la Recherche (ANR-09-BIOT-018) to ET, JMZ, PR and JPP. JG was supported by the Ministerio de Economía y Competitividad (ERA-NET NEURON PCIN-2013–018 C03-02 and SAF2014-58396-R). This project has received funding from the European Union's Seventh Framework Program for research, technological development and demonstration under grant agreement no 627227 (TCM)

## Additional information

### Competing interests

JMZ: Dr Zwier is working at CisBio Bioassays, a company selling HTRF tools. ET: Head of R&D at CisBio Bioassays. The other authors declare that no competing interests exist.

### Funding

| Funder | Grant reference number | Author |
| --- | --- | --- |
| Seventh Framework Programme | 627227 | Thor C Møller |
| European Commission | ERA-NET Neuron PCIN-2013-018-C03-02 | Jesús Giraldo |
| European Commission | ERA-NET Neuron SAF2014-58396-R | Jesús Giraldo |
| Agence Nationale de la Recherche | ANR-09-BIOT-018 | Jurriaan M Zwier<br>Eric Trinquet<br>Philippe Rondard<br>Jean-Philippe Pin |
| Fondation pour la Recherche Médicale | DEQ20130326522 | Jean-Philippe Pin |
| CisBio Bioassays | Laboratoire Collaboratif | Jean-Philippe Pin |

The funders had no role in study design, data collection and interpretation, or the decision to submit the work for publication.

### Author contributions

DMD, Conceptualization, Data curation, Formal analysis, Investigation, Methodology, Writing—original draft; TCM, Formal analysis, Investigation, Methodology, performed the TR-FRET microscopy experiments in cultured neurons, analyzed the results, prepared the figures and text related to these experiments; JS, Conceptualization, Formal analysis, Investigation, Methodology, Writing—review and editing, performed the electrophysiological experiments and wrote the corresponding part of the ms; JG, Conceptualization, Formal analysis, Funding acquisition, Investigation, supervised the mathematical modeling, and wrote the appendix; DM, Conceptualization, Supervision, Methodology, developed the XCelligence method in the laboratory, validating the assay with the proper

controls, and trained D Moreno in setting up the assay for his experiments; XR, Investigation, Methodology; PS, Supervision, Investigation, Methodology, Set up and validated the heterodimer selective sensor assay and trained D Moreno in using this assay. Performed the preliminary experiments at the origin of this project; JMZ, Formal analysis, Methodology, Writing—review and editing, Essential role in the development of the TR-FRET microscope, and supervise the development of the sensor assay. Provided his knowledge in FRET; JP, Conceptualization, Supervision, Writing—review and editing, suêrvised the electrophysiological experiments; TD, Supervision, Methodology, Developed the TR-FRET microscope and trained T Moeller in using it. Participated in the analysis of the TR-FRET microscopy data; ET, Conceptualization, Resources, Supervision, Methodology, Supervise the Cisbio team, and provided a lot of chemical tools essential to the project. Partcipated in the development of the TR-FRET sensor development; LP, Formal analysis, Funding acquisition, Visualization, Methodology, Supervise the arpege facility and initiated the Xcelligence technology in the laboratory. Raised the funding necessary for the acquisition of this technology; PR, Conceptualization, Formal analysis, Supervision, Writing—review and editing, supervise the preliminary experiments, and the TR-FRET sensor development; J-PP, Conceptualization, Supervision, Funding acquisition, Methodology, Writing—review and editing

## Author ORCIDs
David Moreno Delgado, http://orcid.org/0000-0003-0070-2566
Thor C Møller, http://orcid.org/0000-0002-2256-577X
Jesús Giraldo, http://orcid.org/0000-0001-7082-4695
Xavier Rovira, http://orcid.org/0000-0002-9764-9927
Jurriaan M Zwier, http://orcid.org/0000-0001-9017-5856
Philippe Rondard, http://orcid.org/0000-0003-1134-2738
Jean-Philippe Pin, http://orcid.org/0000-0002-1423-345X

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

## Appendix 1

# Analysis of the functional response of the mGlu2-4 system to LY354740 under fixed concentrations of L-AP4

LY354740 is an mGlu2 agonist

L-AP4 is an mGlu4 agonist

Note: LY354740 will be named LY35 for simplification

## Concentration-effect results

Concentration-effect results from our experiments were selected for further analysis with a mathematical model.

## The model

### Model 1

We consider strict mGlu2-4 heterodimers. As we are considering agonists we will focus on the ECDs of the dimeric receptor. Two states either open (O) or closed (C) can be reached by each of the ECDs which can lead to heterodimers arranged as OO, OC or CC dimeric states. We will consider that if all heterodimers were in OO then a functional effect (F) value of 100 would be obtained whereas if they were all in CC an F value of 0 would be observed. Consistently, we assume that OC yields an intermediate value between 0 and 100.

## Assumption

LY35 binds exclusively the mGlu2 protomer whereas L-AP4 binds exclusively the mGlu4 protomer.

With

$$K_1 = \frac{[O^2O^4][A]}{[O_A^2O^4]}; X_1 = \frac{[C^2O^4]}{[O^2O^4]}; X_2 = \frac{[O^2C^4]}{[O^2O^4]}; X_3 = \frac{[C^2C^4]}{[C^2O^4]}; X_4 = \frac{[C^2C^4]}{[O^2C^4]};$$

$$Y_1 = \frac{[C_A^2O^4]}{[O_A^2O^4]}; Y_2 = \frac{[O_A^2C^4]}{[O_A^2O^4]}; Y_3 = \frac{[C_A^2C^4]}{[C_A^2O^4]}; Y_4 = \frac{[C_A^2C^4]}{[O_A^2C^4]}$$

We define the functional response F as

$$F(\%) = \frac{100\left(O^2O^4 + O_A^2O^4 + f\left(C^2O^4 + O^2C^4 + C_A^2O^4 + O_A^2C^4\right)\right)}{[R_T]} \tag{1}$$

With 0 < f < 1.

We consider that OO states produce 100% F, CC states produce 0% F and OC states produce 0 < F(%)<100.

$$F(\%) = \frac{100\left(1 + \frac{[A]}{K_1} + f\left(X_1 + X_2 + (Y_1 + Y_2)\frac{[A]}{K_1}\right)\right)}{1 + X_1 + X_2 + X_1X_3 + (1 + Y_1 + Y_2 + Y_1Y_3)\frac{[A]}{K_1}} \qquad (2)$$

*Equation 2* can be rearranged as the empirical *Equation 3*

$$F(\%) = 100\frac{a + b[A]}{c + d[A]} \qquad (3)$$

With

$$
\begin{aligned}
a &= 1 + f(X_1 + X_2) \\
b &= \frac{1}{K_1}(1 + f(Y_1 + Y_2)) \\
c &= 1 + X_1 + X_2 + X_1X_3 \\
d &= \frac{1}{K_1}(1 + Y_1 + Y_2 + Y_1Y_3)
\end{aligned}
$$

If we divide the numerator and denominator of *Equation 3* by d we have

$$F(\%) = 100\frac{a_1 + a_2[A]}{a_3 + [A]} \qquad (4)$$

It can be shown that *Equation 4* can be written as the typical Hill equation with a Hill coefficient of one (*Equation 5*).

$$F(\%) = \text{Bottom} + \frac{\text{Top} - \text{Bottom}}{1 + 10^{x - x_{50}}} \qquad (5)$$

With $x = \log[A], \text{Bottom} = 100a_2, \text{Top} = 100\frac{a_1}{a_3}, x_{50} = \log a_3$

If we retake the mechanistic constants that define $a_1$, $a_2$, and $a_3$ we can see that

- The basal response (when [A] = 0) is defined by $\text{Top} = 100\frac{a_1}{a_3} = 100\frac{a}{c} = 100\frac{1 + f(X_1 + X_2)}{1 + X_1 + X_2 + X_1X_3}$. Consistently with basal definition there is no constant related with the agonist A.
- The minimum response, that is the asymptotic response as [A] increases, is defined by $\text{Bottom} = 100a_2 = 100\frac{b}{d} = 100\frac{1 + f(Y_1 + Y_2)}{1 + Y_1 + Y_2 + Y_1Y_3}$. Bottom determines the efficacy of the ligand. . . Consequently, the dissociation constant for binding is not present. A full agonist mGlu2 in the heterodimeric context would be one with a high $Y_3$, which leads to the formation of $C^2_A C^4$, that is both protomers are closed. Obviously, if we perform the concentration-response curve of the mGlu2 agonist in the presence of an mGlu4 agonist the closing of the mGlu4 subunit is facilitated, which affects both Top and Bottom.
- The location of the curve along the X=log[A] axis is defined by $x_{50} = \log a_3 = \log\frac{c}{d} = \log\frac{1 + X_1 + X_2 + X_1X_3}{\frac{1}{K_1}(1 + Y_1 + Y_2 + Y_1Y_3)}$. Consistently with potency definition, values related with efficacy (Y constants) and affinity ($K_1$ constant) are present.

Finally, from the slope parameter point we conclude that the proposed mechanistic model with LY35 binding exclusively to mGlu2 and L-AP4 binding exclusively to mGlu4 produce Hill curves with Hill coefficients of one.

## Note

The model can be used for the function of an agonist mGlu4 in the presence of fixed concentrations of an mGlu2 agonist.

Moreno Delgado *et al*. eLife 2017;6:e25233. DOI: 10.7554/eLife.25233

## Data analysis

Experimental data curves were fitted with Hill equations with nH = 1 and with nH allowed to be different from 1. To assess whether nH is statistically different from one different tests can be done.

Data are fitted with the Hill equation $F(\%) = \text{Bottom} + \frac{\text{Top}-\text{Bottom}}{1+10^{nH(x-x_{50})}}$

**Appendix 1—table 1.** Hill equation parameters resulting of fitting curve data in *Appendix 1—figure 1*.

| L-AP4 conc (n) | Top Mean ± SEM | Bottom Mean ± SEM | $x_{50}$ Mean ± SEM | nH Mean ± SEM |
|---|---|---|---|---|
| 0 (3) | 102.72 ± 6.89 | 39.40 ± 2.63 | −7.14 ± 0.11 | 0.77 ± 0.13 |
| −10 (3) | 100.29 ± 1.07 | 47.56 ± 1.29 | −7.40 ± 0.09 | 0.65 ± 0.04 |
| −9 (3) | 104.33 ± 2.04 | 39.05 ± 1.57 | −7.10 ± 0.14 | 0.68 ± 0.10 |
| −8 (3) | 88.64 ± 2.48 | 29.41 ± 1.71 | −7.82 ± 0.11 | 0.67 ± 0.04 |
| −7 (3) | 82.23 ± 2.52 | 29.34 ± 0.62 | −8.79 ± 0.17 | 0.93 ± 0.09 |
| −6.5 (2) | 61.65 ± 3.27 | 24.35 ± 0.74 | −8.21 ± 0.05 | 1.40 ± 0.03 |
| −6 (2) | 54.20 ± 0.48 | 27.81 ± 3.12 | −8.38 ± 0.23 | 1.32 ± 0.11 |
| −5 (2) | 46.14 ± 3.95 | 20.47 ± 3.17 | −8.44 ± 0.20 | 1.21 ± 0.48 |

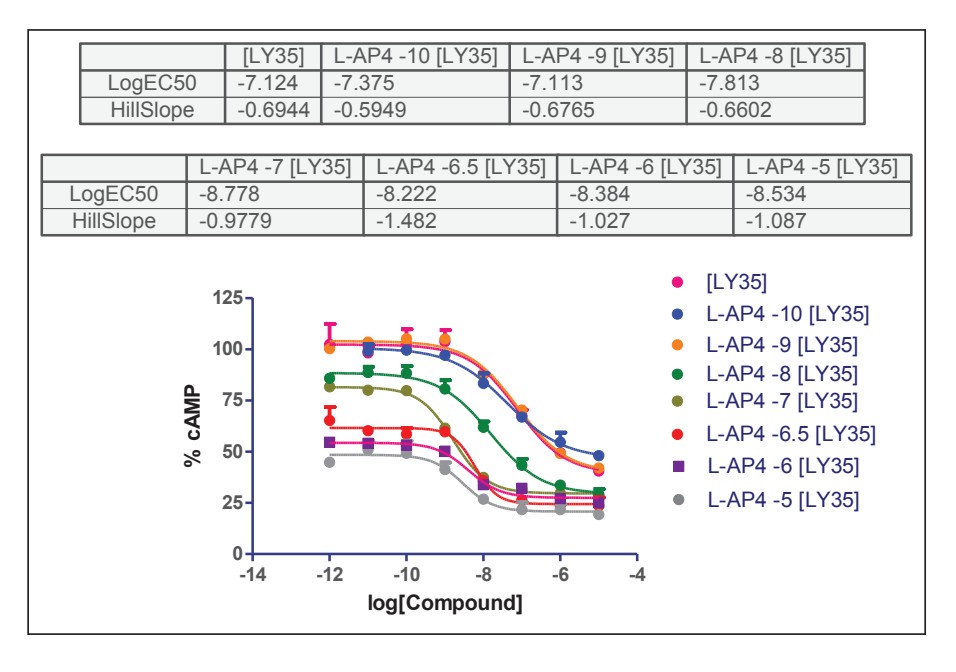

**Appendix 1—figure 1.** Concentration-effect curves of LY35 in the presence of L-AP4.

Addition of mGlu4 agonist displaces the curves downwards because closing of the mGlu4 protomer decreases F. Interestingly an apparent influence on the slope of the curves is observed: The Hill coefficient is less than one when L.AP4 is absent or at low concentration and increases to one at higher L-AP4 concentration.

## Statistical analysis of the Hill coefficient

We analyze whether nH is statistically different from one in two ways.

1.We fit each of the curves with equation $F(\%) = \text{Bottom} + \frac{\text{Top}-\text{Bottom}}{1+10^{\text{nH}(x-x_{50})}}$ and calculate the confidence interval at 95% for m parameter.

**Appendix 1—table 2.** Hill equation parameters resulting of fitting curve data in *Appendix 1—figure 1*.

| L-AP4 conc (n) | nH Mean $\pm$ SEM | Confidence interval of nH parameter (95%) |
|---|---|---|
| 0 (3) | 0.77 $\pm$ 0.13 | (0.21, 1.33) |
| −10 (3) | 0.65 $\pm$ 0.04 | (0.46, 0.83) |
| −9 (3) | 0.68 $\pm$ 0.10 | (0.26, 1.09) |
| −8 (3) | 0.67 $\pm$ 0.04 | (0.49, 0.85) |
| −7 (3) | 0.93 $\pm$ 0.09 | (0.56, 1.30) |
| −6.5 (2) | 1.40 $\pm$ 0.03 | (1.04, 1.75) |
| −6 (2) | 1.32 $\pm$ 0.11 | (−0.12, 2.76) |
| −5 (2) | 1.21 $\pm$ 0.48 | (−4.94, 7.37) |

Curves with $[\text{L-AP4}] \leq 10^{-7}$ present nH <1 in average with some of them ($10^{-10}$, $10^{-8}$) reaching statistical significance (the confidence interval of the nH parameter is below 1) and one of them ($10^{-9}$) very close to be statistically significant.

2.We fit the collection of curves (3 or 2) for each L-AP4 concentration with two equations $F(\%) = \text{Bottom} + \frac{\text{Top}-\text{Bottom}}{1+10^{\text{nH}(x-x_{50})}}$ and $F(\%) = \text{Bottom} + \frac{\text{Top}-\text{Bottom}}{1+10^{x-x_{50}}}$ and analyze with an F-test of the sum of squares errors whether the model including the nH parameter provides a better fit than that in which nH is not present.

**Appendix 1—table 3.** Statistical comparison of goodness of fit including the slope parameter (nH) or not to curves displayed in *Appendix 1—figure 1*.

| [LAP4] | SS1 | df1 | SS2 | df2 | F-value | p-value |
|---|---|---|---|---|---|---|
| 0 | 1430.9 | 21 | 1330.1 | 20 | 1.515675513 | 0.232557 |
| 10**(−10) | 541.3 | 17 | 415.4 | 16 | 4.849301878 | 0.042671 |
| 10**(−9) | 852.8 | 21 | 727.1 | 20 | 3.457571173 | 0.077738 |
| 10**(−8) | 560.5 | 21 | 455.6 | 20 | 4.604916594 | 0.044334 |
| 10**(−7) | 273.4 | 21 | 273.1 | 20 | 0.021969974 | 0.883651 |
| 10**(−6.5) | 220.5 | 12 | 207.8 | 11 | 0.672281039 | 0.429666 |
| 10**(−6) | 206.5 | 13 | 206.5 | 12 | 0 | 1 |
| 10**(−5) | 252.3 | 13 | 252 | 12 | 0.014285714 | 0.906839 |

Results are consistent with the previous analysis, the model including the slope parameter improves significantly the fitting for $[\text{LAP4}] = 10^{-10}$ and $10^{-8}$ and close to significance for $[\text{LAP4}] = 10^{-9}$.

## Comments

The mechanistic model depicted in *Appendix 1—figure 2* yields an empirical Hill equation with a Hill coefficient of 1. Experimental data suggest that the binding of LY35 to the

heterodimer produces curves with Hill coefficient lower than one at low [L-AP4] and curves with Hill coefficient not different from one at high [L-AP4]. Thus, apparently, there is a contradiction between the mechanistic model and those results with the slope parameter lower than one.

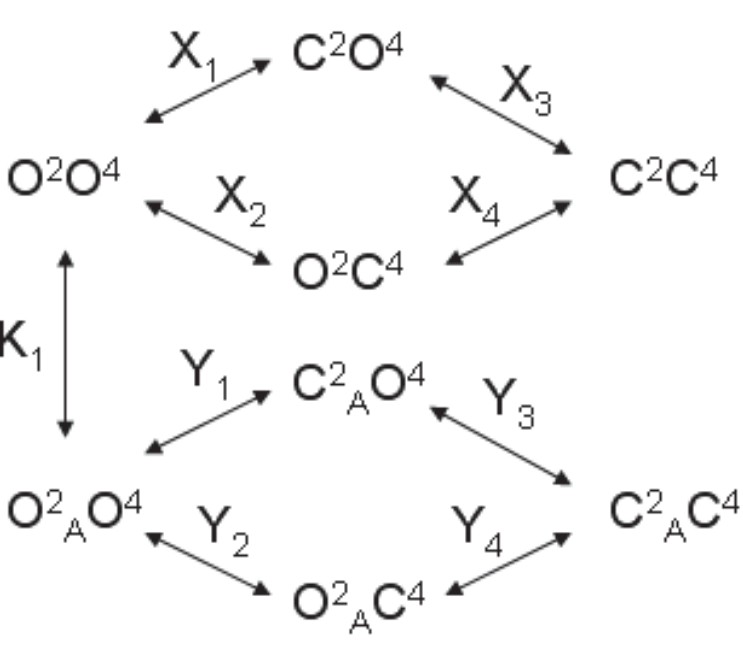

**Appendix 1—figure 2.** Model 1. A heterodimeric mGlu2/4 model in which an mGlu2 agonist binds exclusively the mGlu2 protomer. The binding of an mGlu4 ligand to mGlu4 alters the constants of the model.

A slope parameter lower than one could be explained assuming that LY35 binds at both mGlu2 and mGlu4 subunits in the heterodimer with crosstalk between them. Addition of mGlu4 agonist L-AP4 precludes the binding of LY35 to mGlu4 subunit and converts the heterodimeric receptor in a monomeric receptor for LY35.

# Extending the model

## Model 2

To account for concentration-effect curves with a Hill coefficient different from one the model displayed in *Appendix 1—figure 2* was extended by allowing the possibility that LY35 could bind the mGlu4 protomer in addition to the mGlu2 one (*Appendix 1—figure 3*).

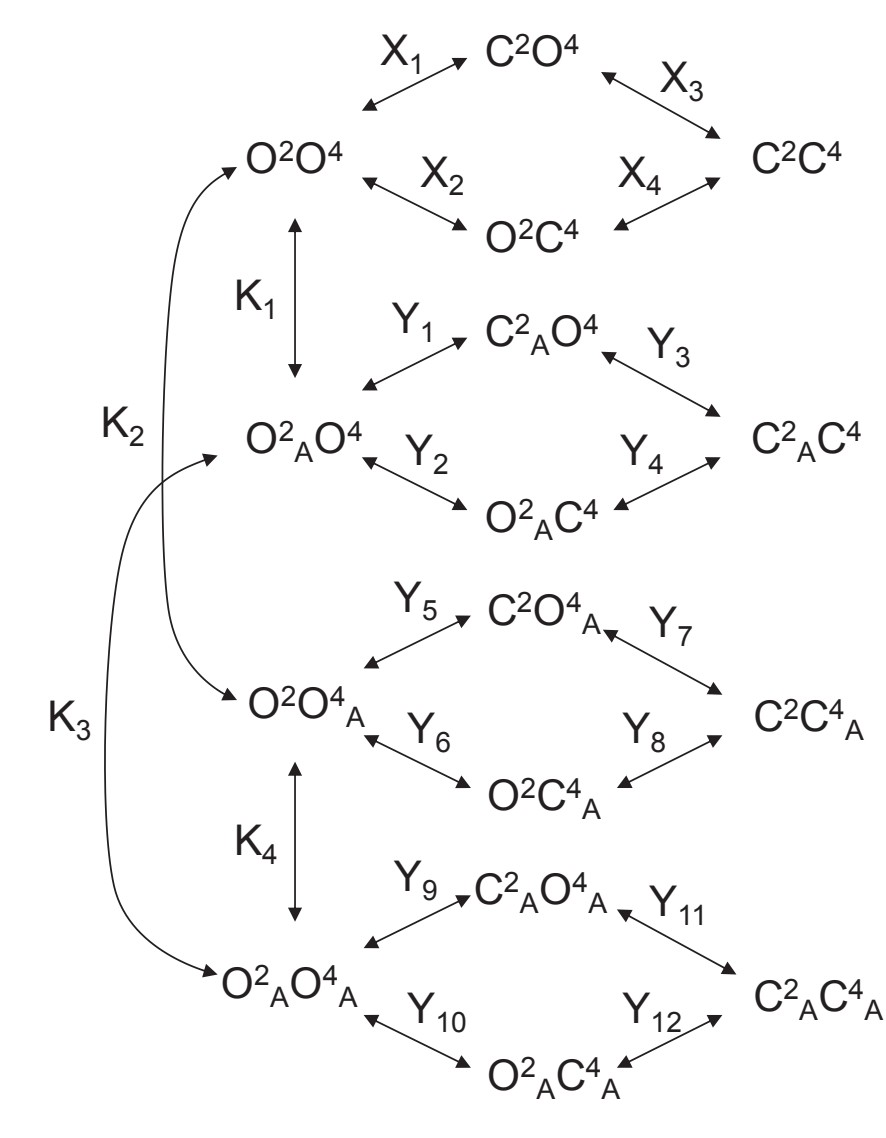

**Appendix 1—figure 3.** Model 2. A heterodimeric mGlu2/4 model in which an mGlu2 agonist binds both the mGlu2 and the mGlu4 protomers. The additional binding of an mGlu4 ligand alters the constants of the model.

With

$$K_1 = \frac{[O^2O^4][A]}{[O_A^2O^4]}; K_2 = \frac{[O^2O^4][A]}{[O^2O_A^4]}; K_3 = \frac{[O_A^2O^4][A]}{[O_A^2O_A^4]}; K_4 = \frac{[O^2O_A^4][A]}{[O_A^2O_A^4]};$$

$$X_1 = \frac{[C^2O^4]}{[O^2O^4]}; X_2 = \frac{[O^2C^4]}{[O^2O^4]}; X_3 = \frac{[C^2C^4]}{[C^2O^4]}; X_4 = \frac{[C^2C^4]}{[O^2C^4]};$$

$$Y_1 = \frac{[C_A^2O^4]}{[O_A^2O^4]}; Y_2 = \frac{[O_A^2C^4]}{[O_A^2O^4]}; Y_3 = \frac{[C_A^2C^4]}{[C_A^2O^4]}; Y_4 = \frac{[C_A^2C^4]}{[O_A^2C^4]};$$

$$Y_5 = \frac{[C^2O_A^4]}{[O^2O_A^4]}; Y_6 = \frac{[O^2C_A^4]}{[O^2O_A^4]}; Y_7 = \frac{[O^2C_A^4]}{[O^2O_A^4]}; Y_8 = \frac{[O^2C_A^4]}{[O^2O_A^4]};$$

$$Y_9 = \frac{[C_A^2O_A^4]}{[O_A^2O_A^4]}; Y_{10} = \frac{[C_A^2O_A^4]}{[O_A^2O_A^4]}; Y_{11} = \frac{[C_A^2O_A^4]}{[O_A^2O_A^4]}; Y_{12} = \frac{[C_A^2O_A^4]}{[O_A^2O_A^4]};$$

We define the functional response F as

$$F(\%) = \frac{100\left(O^2O^4 + O_A^2O^4 + O^2O_A^4 + O_A^2O_A^4 + f\left(\begin{array}{c} C^2O^4 + O^2C^4 + C_A^2O^4 + O_A^2C^4 \\ +C^2O_A^4 + O^2C_A^4 + C_A^2O_A^4 + O_A^2C_A^4 + \end{array}\right)\right)}{[R_T]} \quad (6)$$

With 0 < f < 1.

We consider that OO states produce 100% F, CC states produce 0% F and OC states produce 0 < F(%)<100.

$$F(\%) = \frac{100\left(\begin{array}{c} 1 + f(X_1 + X_2) + [A]\left(\frac{1}{K_1} + \frac{1}{K_2} + f\left(\frac{Y_1+Y_2}{K_1} + \frac{Y_5+Y_6}{K_2}\right)\right) \\ +[A]^2\frac{1}{K_1K_3}(1 + fY_9(1 + Y_{10})) \end{array}\right)}{1 + X_1 + X_2 + X_1X_3 + [A]\left(\frac{1}{K_1} + \frac{1}{K_2} + \frac{Y_1+Y_2}{K_1} + \frac{Y_5+Y_6}{K_2}\right) + [A]^2\frac{1}{K_1K_3}(1 + Y_9(1 + Y_{10}))} \quad (7)$$

*Equation 7* can be rearranged as the empirical *Equation 8*

$$F(\%) = 100\frac{c_1 + c_2[A] + c_3[A]^2}{c_4 + c_5[A] + c_6[A]^2} \quad (8)$$

With

$$\begin{array}{rl} c_1 = & 1 + f(X_1 + X_2) \\ c_2 = & \frac{1}{K_1} + \frac{1}{K_2} + f\left(\frac{Y_1+Y_2}{K_1} + \frac{Y_5+Y_6}{K_2}\right) \\ c_3 = & \frac{1}{K_1K_3}(1 + f(Y_9 + Y_{10})) \\ c_4 = & 1 + X_1 + X_2 + X_1X_3 \\ c_5 = & \frac{1}{K_1} + \frac{1}{K_2} + \frac{Y_1+Y_2+Y_1Y_3}{K_1} + \frac{Y_5+Y_6+Y_5Y_7}{K_2} \\ c_6 = & \frac{1}{K_1K_3}(1 + Y_9 + Y_{10} + Y_9Y_{11}) \end{array}$$

The empirical $c_i$ coefficients reflect, when expressed as combination of mechanistic constants, what we would expect from the comparison between *Equation 8* and *Appendix 1—figure 3*. That is, c1 and c4 include equilibrium constants related with free receptor species only. Analogously, c2 and c5 include equilibrium constants related with receptor species with only one bound agonist whereas c3 and c6 include equilibrium constants related with receptor species with two bound agonists.

Of note, the relationships between the empirical coefficients determine the shape of the concentration-effect curves (see below for shape quantification).

If we divide the numerator and denominator of *Equation 8* by $c_6$ we have *Equation 9*.

$$F(\%) = 100 \frac{a_1 + a_2[A] + a_3[A]^2}{a_4 + a_5[A] + [A]^2} \tag{9}$$

With $a_i = c_i/c_6$ for i = 1 to 6.

*Equation 9* is an empirical equation for a ligand that binds to two receptor sites. This equation has been previously derived from some mechanistic models involving two receptor binding sites (see [1,2] as examples and reviews [3,4]).

The shape of the concentration-effect curve determined by *Equation 9* can be quantitatively characterized by the following geometric determinants.

- The basal response (when [A] = 0) is defined by

$$\text{Top} = 100 \frac{a_1}{a_4} = 100 \frac{c_1}{c_4} = 100 \frac{1 + f(X_1 + X_2)}{1 + X_1 + X_2 + X_1 X_3} \tag{10}$$

Consistently with basal definition there is no constant related with the agonist A.

Because 0<f < 1, Top will be always lower than 100. Furthermore, as higher is X3 (the propensity to form CC states) lower is Top.

- The minimum response, that is the asymptotic response as [A] increases, is defined by

$$\text{Bottom} = 100 a_3 = 100 \frac{c_3}{c_6} = 100 \frac{1 + f(Y_9 + Y_{10})}{1 + Y_9 + Y_{10} + Y_9 Y_{11}} \tag{11}$$

Bottom determines the efficacy of the ligand. Considering the mechanistic constants included in Bottom definition, it follows that a3 <1. A ligand is a full agonist if $a_3 << 1$ and a partial agonist if $a_3 < 1$.

In agreement with Bottom defined as efficacy, the dissociation constants for binding are not present in its mechanistic expression. A full agonist mGlu2 in the heterodimeric context would be one with a high $Y_{11}$, which leads to the formation of $C^2_A C^4_A$, that is, both protomers are closed. Obviously, if we perform the concentration-response curve of the mGlu2 agonist in the presence of an mGlu4 agonist the closing of the mGlu4 subunit is facilitated, which affects both Top and Bottom.

- The location of the curve along the X=log[A] axis ($X_{50}$=log[$A_{50}$] or mid-point) defines the potency of the ligand and is defined by

$$X_{50} = \log\left(\frac{-b \pm \sqrt{b^2 - 4ac}}{2a}\right) \tag{12}$$

Consistently with potency definition, values related with efficacy (Y constants) and affinity ($K_i$ constants) are present.

Where

$$a = a_1 - a_3 a_4; \; a = a_3 a_4 a_5 - 2a_2 a_4 + a_1 a_5; \; \text{and} \; c = -a_4(a_1(a_1 - a_3 a_4))$$

- •Quantification of cooperativity by the calculation of the Hill coefficient can be done by making use of the definition of the Hill coefficient at the mid-point ($n_{H_{50}}$) for a given y(x) function.[5]

$$n_{H_{50}} = \frac{4\left(\frac{dy}{dx}\right)_{x_{50}}}{a \ln 10} \tag{13}$$

With y=F(%), x = log[A]; a, the Bottom; ln, the natural logarithm; and d/dx, the derivative operator as expressed in *Equation 14*.

$$\frac{dy}{dx} = \frac{100\left(-\left(a_1 + a_2 10^x + a_3 10^{2x}\right)\left(a_5 10^x + 2\cdot 10^{2x}\right) + \left(2a_3 10^{2x} + a_2 10^x\right)\left(a_4 + a_5 10^x + 10^{2x}\right)\right)\ln 10}{\left(a_4 + a_5 10^x + 10^{2x}\right)^2} \quad (14)$$

The value of the Hill coefficient as obtained from *Equation 13*, with empirical coefficients ($a_1$ to $a_5$), which in turn are defined in terms of mechanistic equilibrium constants, may provide a mechanistic interpretation to the Hill coefficient obtained by fitting with the empirical Hill equation including the slope parameter.

The ratio $c_1/c_4$ determines the Top asymptote (basal response) and the ratio $c_3/c_6$ determines the Bottom asymptote (efficacy). The ratio $c_2/c_5$ determines the sensitivity of the measured effect to agonist concentration. Considering the mechanistic constants included in $c_2$ and $c_5$ definition, it follows that $c_2 < c_5$. The induction constants that appear in c2 and c5 expressions are $Y_1$, $Y_2$, $Y_3$, $Y_5$, $Y_6$ and $Y_7$, which are those constants affecting receptor species with only one molecule of mGlu2 agonist present. We see that $Y_3$ and $Y_7$ are present in $c_5$ but not in $c_2$; then, the values of these constants may modulate the $c_2/c_5$ ratio. Because of the closure of the thermodynamic cycles included in the model displayed in Fig. S3, $Y_4$ and $Y_8$ ($Y_4 = \frac{Y_1}{Y_2}Y_3$, $Y_8 = \frac{Y_5}{Y_6}Y_7$) can be used instead of $Y_3$ and $Y_7$, respectively. The pair ($Y_3$, $Y_7$) or the pair ($Y_4$, $Y_8$) measure how the closure of one protomer favors the closure of the other thus it can be considered as a measure of functional cooperativity. Thus, we can conclude that the functional cooperativity between the two protomers affects the sensitivity of the measured effect and be the cause of some of the flat curves observed.

*Equation 9* contains five parameters and is difficult to fit to curves that do not display a clear biphasic shape. However it may be used for modeling different pharmacological conditions by assigning particular values to the parameters.

## Simulation of pharmacological conditions under the mechanistic models

*Appendix 1—figure 4* illustrates how Model two can explain the flat curve observed for LY35 mGlu2 agonist. We assume that the closed-closed state is not achieved by proposing the induction constants $Y_3 = Y_7 = Y_{11} = 10^{-6}$. These constants make LY35 to behave as a partial agonist with a bottom value of 51%. We assume that there is negative binding cooperativity and the ligand binds the mGlu4 protomer after occupying first the mGlu2 binding site ($K_1 = 10^{-6}$; $K_2 = 10^3$; $K_3 = 10^{-3}$). This leads to a curve with two components, one related with the binding of the first molecule to the heterodimer and another one related with the binding of the second molecule. The induction constant for the closure of the mGlu2 subunit is greater in the doubly- than in the singly-bound heterodimer ($Y_9 > Y_1$). A f-value of 0.5 was used for the functional closed-open state.

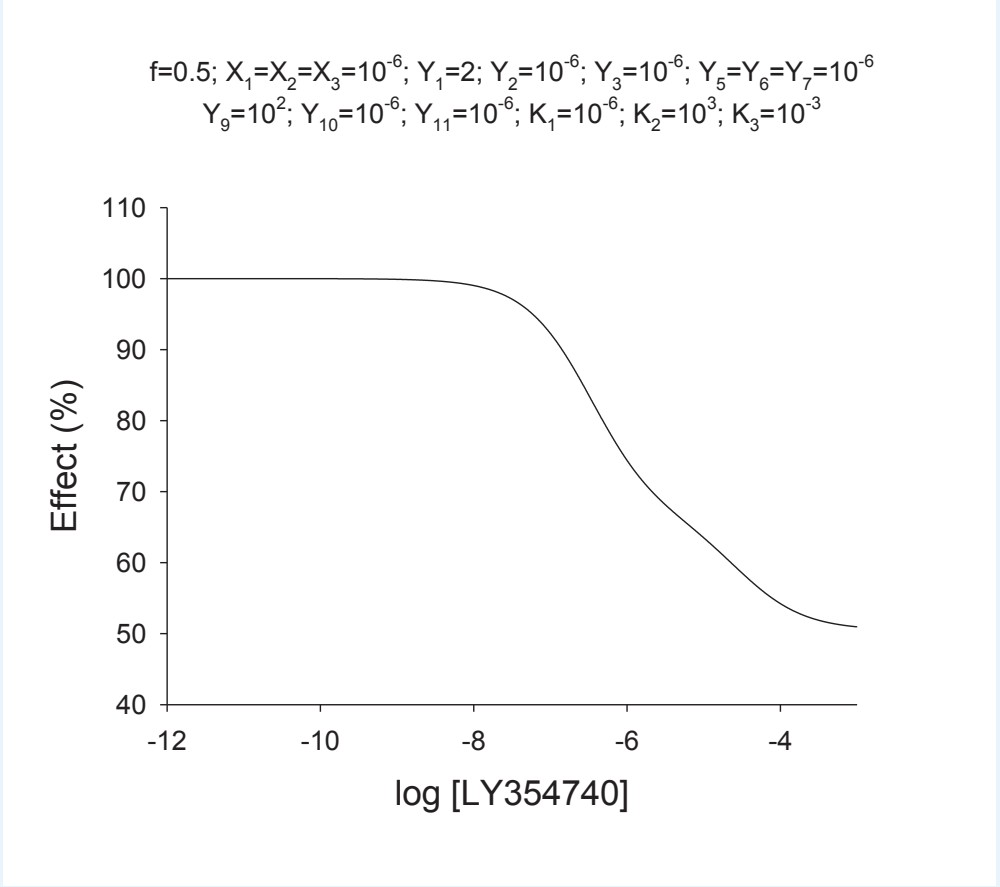

**Appendix 1—figure 4.** Theoretical concentration-effect curve for particular values of the mechanistic constants included in Model 2. The theoretical concentration-effect data are the following.

Fitting with the Hill equation $F(\%) = \text{Bottom} + \frac{\text{Top}-\text{Bottom}}{1+10^{\text{nH}(x-x_{50})}}$ yielded the following parameters

**Appendix 1—table 4.** Data extracted from concentration-effect curve of *Appendix 1—figure 4*.

| Log[LY354740] | Effect (%) |
|---|---|
| −12.00 | 100.00 |
| −11.00 | 100.00 |
| −10.00 | 99.99 |
| −9.00 | 99.90 |
| −8.00 | 99.03 |
| −7.00 | 92.28 |
| −6.00 | 74.40 |
| −5.00 | 63.50 |
| −4.00 | 54.23 |
| −3.00 | 50.96 |

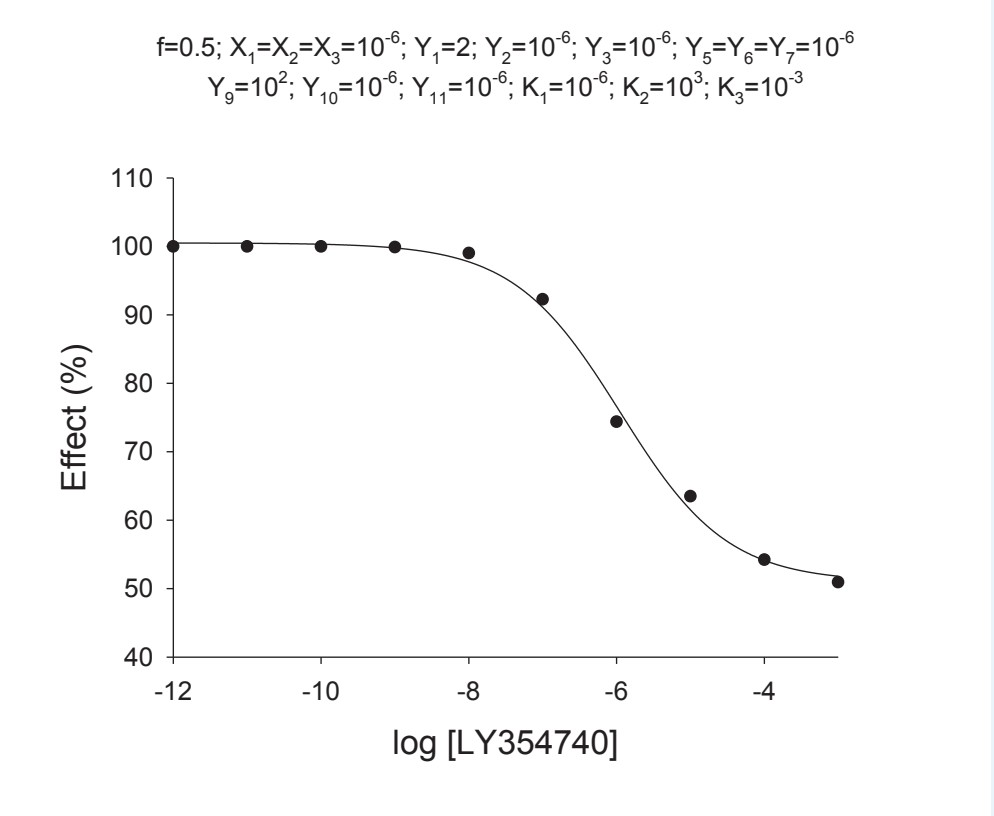

$f=0.5$; $X_1=X_2=X_3=10^{-6}$; $Y_1=2$; $Y_2=10^{-6}$; $Y_3=10^{-6}$; $Y_5=Y_6=Y_7=10^{-6}$
$Y_9=10^2$; $Y_{10}=10^{-6}$; $Y_{11}=10^{-6}$; $K_1=10^{-6}$; $K_2=10^3$; $K_3=10^{-3}$

**Appendix 1—figure 5.** Curve data included in *Appendix 1—table 4* (solid points) and the theoretical curve by using the Hill equation parameters of *Appendix 1—table 5* (curve line).

The Hill coefficient nH is ~0.6 in agreement with experimental data.

The following graph includes the theoretical data from Model two and the curve produced by using the Hill equation fitted parameters.

**Appendix—table 5.** Parameter values by fitting curve data in *Appendix 1—table 4* with the Hill equation.

| Parameter | Estimate | Approx std error | Approximate 95% Confidence Limits | |
|---|---|---|---|---|
| **Bottom** | **50.9874** | **1.5333** | **47.2355** | **54.7393** |
| Top | 100.5 | 0.7699 | 98.6649 | 102.4 |
| x50 | −5.9487 | 0.0920 | −6.1739 | −5.7234 |
| nH | 0.5993 | 0.0669 | 0.4356 | 0.7631 |

