## [Decision Letter]

Thank you for submitting your article "Pharmacological evidence for a metabotropic glutamate receptor heterodimer in neurons" for consideration by *eLife*. Your article has been reviewed by three peer reviewers, one of whom is a member of our Board of Reviewing Editors, and the evaluation has been overseen by Richard Aldrich as the Senior Editor. The following individuals involved in review of your submission have agreed to reveal their identity: Robert Duviosin (Reviewer #3).

The reviewers have discussed the reviews with one another and the Reviewing Editor has drafted this decision to help you prepare a revised submission.

The manuscript was evaluated by knowledgeable referees, who provided positive comments about the findings of a mGluR2-4 heterodimer. Using a time resolved FRET based approach, strong evidence was provided for a heterodimeric arrangement of mGlu2-4 in HEK293 cells and primary neurons after transfection experiments. Agonists of mGlu2 were used to define the responses and cooperativity of the Glu2-4 heterodimer with a determination of the Hill coefficient. Several reservations were raised, including (1) the relevance of the findings to the endogenous receptor; (2) the need for additional data in neuronal systems where mGlu2-4 receptors are found; and (3) a number experimental details need to be clarified and rectified throughout the manuscript. It was felt that the Background and Materials and methods require more explanation, since the methods represent a new way of demonstrating heteromer interactions. We feel that the majority of these issues can be addressable in a timely fashion and recommend a major revision for this paper. The major concerns are listed below.

Review #1:

Although the analysis is compelling, the study would be strengthened if the transfection experiments excluded the effects of overexpression and miss localization of the receptors.

1) An expression system is used to follow the mGlu2-4 heterodimers at the cell surface, but this does not exclude an intracellular localization of the receptors. Also, strong promoters were used to gain high expression levels in HEK293 and primary hippocampal neurons. Some attention should be given to appropriate cellular expression after transfection. The images in Figure 5 and Figure 6 are adequate, but do not indicate if there is are differences in subcellular localization after transfection.

2) A SThdHQ7 striatal cell line is used to follow mGlu2-4 receptors. Why do the cells shrink after activation of adenylate cyclase (Results section)? There is not a good explanation for why these cells are chosen. What is the relevance of mGlu2-4 to striatal disorders, such as Huntington's disease?

3) How is "minimum leaking of respective homodimers" determined? A definition of the "orthosteric binding site" of mGluR (subsection “Activation of both subunits in mGlu2-4 receptor is required for full activity**”**) would be helpful.

4) The stoichiometry of many other receptor systems has been deduced from affinity crosslinking experiments. Given all the tools available to the laboratory, this is an accessible experiment and would provide more evidence for the heterodimer model.

Reviewer #2:

In this manuscript the authors examine heterodimerization between metabotropic glutamate receptor types in modulating receptor activity. The majority of the studies are carried out with recombinant receptors- while the studies are solid and address an important question, a cursory effort has been made to characterize the native receptors. The authors use primarily heterologous or neuronal cell lines and FRET-based techniques to explore the changes in movement of the receptors upon ligand binding. They go on to explore the regulation of signaling altered by this process. The studies are well thought out and executed and conclusions are based on solid data. In sum, the majority of the studies are carried out with recombinant receptors expressed in heterologous cells or cultured neurons – while the studies are solid and address an important question in the field, how relevant this information is to the native receptor in an endogenous setting is not well explored.

1) Title – as written the title makes a sweeping claim that is only partially addressed and so is misleading.

2) Abstract – This is written in rather general terms and reads like conclusions rather than the synopsis of the study. Expanding the Abstract to include a bit more information as to the major findings of the study would help.

3) Materials and methods – More details need to be included in this section and some important details are missing; for example, details of the receptors (including species) transfected into neurons is not included here (it appears in the Figure legend). How the ER-retaining receptors were generated, characterized and independently tested so that only heteromers appear at the cell surface is not clear. Authors are encouraged to show this data as it would strengthen their argument that only heteromers are expressed at the cell surface. These are only a couple of examples and the authors are should carefully go over each of the experiments in the manuscript to ensure that all of the methods are clearly described rather than having the reader go back and forth between the Materials and methods and the figure legend section.

Results

1)Subsection “A FRET-based sensor to identify mGlu2-4 specific fingerprints**”**: As written it appears that the concentrations of SNAP-donor and CLIP-acceptor substrates that gave a signal for the heteromer did not give a TR-FRET signal with respective receptor homomers – is this true? This needs to be clarified in the manuscript.

2). In the same subsection, paragraph two: Figure 1 is a schematic representation of how homomers and the heteromer are tagged; this figure does not show the decreases in TR-FRET as it is described in the text. Appropriate panels showing the actual data need to be referenced (not a schematic).

3)A table with potencies, efficacies, Hill coefficient with the different ligands in cells expressing homomers and heteromers would be useful as it would enable the reader to compare changes instead of going through different figures to check out statements made in the text.

4)Figure 7 contains one panel that addresses the physiological relevance of the findings in this study. A combination of agonists (albeit at a single concentration of each) give more than an additive response when compared to the responses to the individual agonists. However, the error bars are high and it is not clear how many cells were examined and from how many animals (this should be in Figure legend). In the figure legend the number of (n) independent experiments not provided.

5) Also a dose response curve of either LY34540 (or LSP4-2022) in the presence of a low dose of LSP4-2022 (or LY34540) would have strengthened this crucial point.

6) Figure 1—figure supplement 1 what do the different colors in the plots mean. How do they help differentiate homomeric from heteromeric signals?

7.)There is a mismatch between the title of the figure and figure legend – the authors are requested to check this carefully. For example, legend on Figure 1—figure supplement 1 says mGlu4 but title of figure says mGlu2, legend for Figure 1—figure supplement 1 says mGlu2 but title of figure says mGlu4. Similarly in Figure 1—figure supplement 2 and B title in the figure says mGlu2 agonist but Results section states mGlu4 agonist.

8) Most importantly, neither the co-localization of the two receptors in this system nor the absence of synergy in animals lacking one of the receptors has been explored.

Reviewer #3:

This manuscript presents very nice data in support of mGlu2-mGlu4 heterodimer receptors. The FRET experiments are convincing and demonstrate that the pharmacology of herodimers can be distinguished from that of either homodimers. Second, the authors show that several compounds have distinct actions on hererodimers compared to homodimers, such as different Hill coefficients or partial agonist or synergistic activities. Finally the experiments measuring cell swelling of cultured striatal neuron cell line as a proxy for adenylate cyclase activation are inovative and reveal the synergistic activity of BINA and VU0415374 in neurons. My only question about this manuscript is that the evidence of mGlu2-4 heterodimers in the lateral perforant path is most suggestive, but it is not conclusive. The synergistic effects shown could be caused downstream of the receptors in the Gi-coupled pathways. Given the pharmacological fingerprints presented in Figure 4 or 6, can similar experiments be done in hippocampal slices?

---

## [Author Response]

The manuscript was evaluated by knowledgeable referees, who provided positive comments about the findings of a mGluR2-4 heterodimer. Using a time resolved FRET based approach, strong evidence was provided for a heterodimeric arrangement of mGlu2-4 in HEK293 cells and primary neurons after transfection experiments. Agonists of mGlu2 were used to define the responses and cooperativity of the Glu2-4 heterodimer with a determination of the Hill coefficient. Several reservations were raised, including;

*1) The relevance of the findings to the endogenous receptor;*

Firstly, our data indicate that the formation of mGlu2-4 heterodimers is not the consequence of an over-expression of these subunits. Of note, the expression levels of both mGlu2 and mGlu4 are rather low in the striatal cell line, such that no direct responses could be measured using conventional second messenger assays. This is only when using a label free assay that mGluR responses could be measured.

Also, our quantification of transfected mGlu2 receptors in hippocampal neurons revealed a 5-fold only increase in cell surface receptors compared to controls.

We also report that the synergistic effect observed in hippocampal slices is lost in the mGlu4 KO mice, and that the increase in potency of the mGlu2 agonist LY354740 observed upon activation of the mGlu4 subunit cannot be reproduced by activating a co-expressed the Gi-coupled δ opioid receptor. We think our study brings strong pharmacological evidence for the existence of mGlu2-4 heterodimers in cells naturally expressing these receptors, and that this phenomenon is not the consequence of receptor over-expression. We agree that more work is needed to decipher the physiological significance of such heteromeric assemblies, but this is not the scope of our present manuscript that mainly concentrated on the development of assays aimed at characterizing mGlu heterodimers.

2) The need for additional data in neuronal systems where mGlu2-4 receptors are found;

Data with mGlu4 KO mice have been added, and confirmed that the LSP4-2022 effect is mediated through the mGlu4 subunit. As expected, no synergistic effect is observed between LY354740 and LSP4-2022 in the absence of mGlu4.

*3) A number experimental details need to be clarified and rectified throughout the manuscript. It was felt that the Background and Materials and methods require more explanation, since the methods represent a new way of demonstrating heteromer interactions. We feel that the majority of these issues can be addressable in a timely fashion and recommend a major revision for this paper. The major concerns are listed below.*

The Materials and methods section has been completed with as much information as possible to allow any reader to repeat the experiments.

*Review #1:*

*Although the analysis is compelling, the study would be strengthened if the transfection experiments excluded the effects of overexpression and miss localization of the receptors.*

*1) An expression system is used to follow the mGlu2-4 heterodimers at the cell surface, but this does not exclude an intracellular localization of the receptors. Also, strong promoters were used to gain high expression levels in HEK293 and primary hippocampal neurons. Some attention should be given to appropriate cellular expression after transfection. The images in Figure 5 and Figure 6 are adequate, but do not indicate if there is are differences in subcellular localization after transfection.*

There are many questions raised in this first point of the referee that we will clarify, one by one.

Regarding the validity of the quality control system used to allow a specific cell surface expression of the mGlu2-4 heterodimers, important information can be found in Figure 1—figure supplement 3. Panel 3A shows the cell surface expression of SNAP-mGlu4-C1 relative to SNAP-mGlu4 as a function of plasmid DNA used in the transfection. Under the conditions used, SNAP-mGlu4-C1 surface expression is 0.4% that of SNAP-mGlu4 under our experimental conditions. Panel B shows that co-expression of SNAPmGlu4-C1 with CLIP-mGlu2-C2 leads to a large increase in cell surface expression of the SNAP-mGlu4-C1. Eventually, panel C illustrates that the very low CLIP-CLIP and SNAP-SNAP TR-FRET signals likely originate from bystander FRET rather than FRET between subunit of homodimeric receptors since no decrease in TR-FRET signal is observed upon receptor activation with a specific agonist, in contrast to what is expected if the FRET originated from a heterodimer. We modified the text to make this point clearer to the readers

The referee is right that either assay (heterodimer specific biosensor, or specific targeting of the heterodimer at the cell surface) are conducted in the presence of intracellular receptors. However, when using the biosensor, because we used non cell permeant CLIP and SNAP substrates, only cell surface receptors carrying an extracellular CLIP or SNAP tags can be labeled. Only those cell surface receptors will then be recorded in any biosensor assays. Regarding the ER retention of the C1 and C2 constructs, although retained intracellularly, and even though some authors argue that intracellular mGluRs can signal, we have never been able to measure any signaling with the C1 or C2 constructs expressed alone upon glutamate application (Huang et al., 2011, Brock et al., 2007). We modified the text to make this point clearer to the readers.

The referee also questions the possibility that the reported data results from an over-expression of the receptors. We don't think this is the case, for the following reasons.

i) The interaction between cell surface receptors may be artificially favored due to an over-expression of the proteins. This is well illustrated by the work by Calebiro et al., 2013 reporting that the proportion of dimers and oligomers increases with receptor density at the cell surface. Under our experimental condition, although we can measure TR-FRET between subunits, we have not been able to measure significant signal under conditions where TR-FRET can only result from the proximity between dimers (Maurel et al., 2008; Doumazane et al., 2011). Moreover, various experimental protocols were used to estimate the size of the mGlu2-4 complexes and all were consistent with strict heterodimers (Doumazane et al., 2011).

ii) The functional data obtained with the striatal cell line (Figure 6) nicely illustrate

that a receptor with the pharmacological properties of the mGlu2-4 heterodimer can be functionally detected whereas these cells express a low amount of each receptor as evidence by the difficulty in measuring a second messenger response upon agonist activation. Of note, these functional data were obtained in control nontransfected cells naturally expressing mGlu2 and mGlu4.

iii) The mGlu2 receptor expression in transfected neurons has been quantified, and is reported in another study to be submitted soon. Because a few percent of the cultured neurons are transfected under our experimental conditions, radioactive ligands could not be used to quantify mGlu2 receptor expression. We then took advantage of an mGlu2 specific nanobody (publication submitted) labeled with a fluorophore to compare the fluorescent signal in non-transfected neurons and in transfected neurons. Data are shown in the Figure 13, and show that the mean expression of mGlu2 receptors is 5 times higher than the basal expression in non-transfected neurons. In the study, we also provide evidence that mGlu2 homodimers, despite the 5 times higher expression, do not form larger complexes under control conditions, remaining strict dimers, again illustrating that the mGlu2-4 association observed in the present study unlikely results from a too high expression of the subunits.

Author response image 1.Overlap between expression levels of mGlu_2_ in native and transfected neurons.Expression of mGlu_2_ receptor in native and transfected primary hippocampal neurons was measured by immunofluorescence using a d2-labeled single domain antibody (DN1-d2) specific for the mGlu_2_ receptor. (**A**) Schematic drawing of DN1-d2 labeling of the mGlu_2_ receptor. (**B**) Example of DN1-d2 labeled native neuron with endogenous expression of mGlu_2_ receptor. (**C**) Mean intensity of DN1-d2 labeling of neurons with endogenous mGlu_2_ receptor expression (28 cells) or transfected with mGlu_2_ receptor (26 cells). The endogenous mGlu_2_ expression overlaps with the expression in transfected neurons in the 55-170 subunits/V_eff_ range (shaded area).**DOI:**
http://dx.doi.org/10.7554/eLife.25233.029

A specific paragraph in the Discussion has been added to discuss this issue.

The last aspect raised in this first point of the referee relates to the subcellular localization of the receptors. As illustrated in Figure 5, the transfected subunits are found in most neuronal compartments, whereas both mGlu2 and mGlu4 receptors are expected to be mainly targeted to the terminals. However, when using our specific mGlu2 nanobody (European Patent Application EP3164415, Scholler et al., under revision), mGlu2 receptors could be identified in a few hippocampal neurons, with a wide distribution of the labeling on the entire neuron (see Figure 14), as observed with the transfected receptors (Figure 5). However, the primary goal of this part of our work was to examine whether mGlu2-4 heterodimers can be detected in cells that endogenously express these subunits. We think our data illustrate that this is indeed the case, with the above-mentioned limitation.

Author response image 2.**DOI:**
http://dx.doi.org/10.7554/eLife.25233.030

Data shown in Figure 6 are from control cells, not from transfected cells, so there is no specific issue here regarding miss localization due to transfection.

*2) A SThdHQ7 striatal cell line is used to follow mGlu2-4 receptors. Why do the cells shrink after activation of adenylate cyclase (Results section)? There is not a good explanation for why these cells are chosen. What is the relevance of mGlu2-4 to striatal disorders, such as Huntington's disease?*

Forskolin induces the shrinking of several cell lines. Indeed, forskolin is one of the classical components of differentiation cocktails (Experimental & Molecular Medicine (2004) 36, 52–56; BMC Cell Biol. 2010 Apr 16;11:25). STHdh, can be differentiated using a cocktail containing forskolin. During our initial studies with these cells, and looking for mGlu2 or mGlu4 inhibition of forskolin-induced cAMP production, we came up to this interesting observation that can be easily followed using the Xcelligence label-free system. We then used this characteristic to evaluate Gi protein activation by mGlu2 and mGlu4 agonists.

STHdh cells are well characterized and largely used as a neuronal model. These cells express different key proteins expressed in striatal neurons (DARPP-32, PSD95, D1, EAAT…). As a search for a neuronal cell line to test the possible formation of mGlu24 heterodimers, we chose these cells as an alternative to cultured neurons as such cells are easier to transfect. Of note, mGlu4 is known to be expressed in some striatal neurons (those projecting to the globus pallidus) (Beurrier et al., 2009). Although mGlu2 receptors are found in the striatum (Wright et al., 2013), these mostly correspond to pre-synaptic receptors in cortico-striatal terminals (Ohishi et al., 1993; Conn et al., 2005) since low in situ hybridization signal could be detected in striatal cells (Ohishi et al., 1993; Gu et al., 2008).

We then checked whether mGlu2 and or mGlu4 mediated responses could be detected in these cells, and it was the case using the sensitive label free assay.

These cells were primarily chosen as a neuronal cell model, rather than a model for striatal neurons with the possibility to propose hypothesis regarding the role of mGlu2-4 heterodimer in the basal ganglia network, and their possible use to treat Huntington or Parkinson diseases. We think a more integrated work with more native situation will be necessary before entering into such discussion.

*3) How is "minimum leaking of respective homodimers" determined? A definition of the "orthosteric binding site" of mGluR (subsection “Activation of both subunits in mGlu2-4 receptor is required for full activity**”**) would be helpful.*

In Figure 1—figure supplement 3, SNAP-mGlu4 and CLIP-mGlu2 levels at the cell surface are determined. No significant amount of CLIP-mGlu2 C2KKXX was observed at the cell surface in agreement with our previous studies (Doumazane et al., 2011; Huang et al., 2011), and as such these data were not included. We now refer to our previous work in the main text. However, expression of 0.4% SNAP-mGlu4C1KKXX was detected in comparison with SNAP-mGlu4 WT, as indicated in Figure 1—figure supplement 3. This was considered a “minimum leaking”.

The orthosteric site is the site were the natural ligand binds, in contrast to the allosteric site, defined as a site different from the orthosteric site (Christopoulos et al., Pharmacol Rev 2014). To clarify the issue, we replaced “orthosteric” by “glutamate” so that non-specialist readers will understand that we mutated the binding site where the natural ligand binds.

*4) The stoichiometry of many other receptor systems has been deduced from affinity crosslinking experiments. Given all the tools available to the laboratory, this is an accessible experiment and would provide more evidence for the heterodimer model.*

Although we agree that crosslinking experiments can be informative, such experiments are also highly criticized because transient association can be sufficient for crosslinking. Accordingly crosslinked species will accumulate during the duration of the crosslinking experiment. A nice example is reported in our recent paper (Xue et al., Nat Chem Biol 2015) where crosslinking could be observed between mGlu2 dimers providing these are in the active form. Such crosslinking resulted in the appearance of FRET between dimers. However, no such FRET could be detected without crosslinking, indicating that only a very small proportion of the dimers were associated into larger complexes under control condition, and that such complexes accumulated upon crosslinking.

To us, the best illustration that mGlu2 and mGlu4 associate into heterodimers, rather than the association between two homodimers, is the large change in FRET upon receptor activation, illustrating that the FRET really originates from a specific association of the subunits as observed with the homodimers. In addition, in our previous study (Doumazane et al., 2011) we used several approaches to demonstrate that mGlu2 and mGlu4 form heterodimers at the cell surface rather than larger complexes. We now refer to this previous study in our revised manuscript.

*Reviewer #2:*

*In this manuscript the authors examine heterodimerization between metabotropic glutamate receptor types in modulating receptor activity. The majority of the studies are carried out with recombinant receptors- while the studies are solid and address an important question, a cursory effort has been made to characterize the native receptors. The authors use primarily heterologous or neuronal cell lines and FRET-based techniques to explore the changes in movement of the receptors upon ligand binding. They go on to explore the regulation of signaling altered by this process. The studies are well thought out and executed and conclusions are based on solid data. In sum, the majority of the studies are carried out with recombinant receptors expressed in heterologous cells or cultured neurons – while the studies are solid and address an important question in the field, how relevant this information is to the native receptor in an endogenous setting is not well explored.*

*1) Title – as written the title makes a sweeping claim that is only partially addressed and so is misleading.*

The title has been modified as follow: "Pharmacological evidence for native metabotropic glutamate receptor heterodimers in neuronal cells"

We still think the data obtained with the neuronal cell line, support the existence of native heterodimeric mGlu2-4 in non-transfected cells. Although our data in hippocampal slices are not fully conclusive, as discussed with referee #3, still they are consistent with our proposal. Indeed, such a strong synergy between Gi-coupled receptor agonists has never been observed through signaling synergism. Because our study was primarily aimed at establishing innovative approaches to specifically study the properties of mGlu heterodimers, and because the identified properties can be observed in cells naturally expressing both mGlu2 and mGlu4, we think our new proposed title is appropriate.

*2) Abstract – This is written in rather general terms and reads like conclusions rather than the synopsis of the study. Expanding the Abstract to include a bit more information as to the major findings of the study would help.*

We still think the information given in the Abstract is necessary for a wide readership. Unfortunately, the Abstract could not be extended due to length constraint.

*3) Materials and methods – More details need to be included in this section and some important details are missing; for example, details of the receptors (including species) transfected into neurons is not included here (it appears in the Figure legend). How the ER-retaining receptors were generated, characterized and independently tested so that only heteromers appear at the cell surface is not clear. Authors are encouraged to show this data as it would strengthen their argument that only heteromers are expressed at the cell surface. These are only a couple of examples and the authors are should carefully go over each of the experiments in the manuscript to ensure that all of the methods are clearly described rather than having the reader go back and forth between the Materials and methods and the figure legend section.*

We are really sorry for not checking carefully the Materials and methods section before submission of our manuscript. We thank the referee for pointing all these missing parts. The Materials and methods section has now been complemented and includes all details the reviewer requested. Controls using ER-retaining constructs can be found in Figure 1—figure supplement 3 as detailed previously.

*Results*

*1) Subsection “A FRET-based sensor to identify mGlu2-4 specific fingerprints**”**: As written it appears that the concentrations of SNAP-donor and CLIP-acceptor substrates that gave a signal for the heteromer did not give a TR-FRET signal with respective receptor homomers – is this true? This needs to be clarified in the manuscript.*

When using as substrates SNAP-donor and CLIP-acceptor, then the homodimers composed of two SNAP-tagged subunits only carries a donor, such that no FRET signal is generated. Similarly, the CLIP-tagged homodimer is labeled with the acceptor only, and will also not generate a FRET signal. Then, as indicated in the text, only the heterodimer composed of one subunit tagged with a SNAP, and another subunit tagged with a CLIP, will be labeled with both a donor and an acceptor, such that TR-FRET signals will be generated by the heterodimer only. This doesn’t mean (as suggested by the referee’s comment) that homomers cannot be formed, but simply that our experimental conditions allow to monitor by FRET signals coming from the heteromers, exclusively.

This point has been made clearer in our revised manuscript.

*2). In the same subsection, paragraph two: Figure 1 is a schematic representation of how homomers and the heteromer are tagged; this figure does not show the decreases in TR-FRET as it is described in the text. Appropriate panels showing the actual data need to be referenced (not a schematic).*

This has been corrected.

*3)A table with potencies, efficacies, Hill coefficient with the different ligands in cells expressing homomers and heteromers would be useful as it would enable the reader to compare changes instead of going through different figures to check out statements made in the text.*

The requested Table has been added (Table 1).

*4)Figure 7 contains one panel that addresses the physiological relevance of the findings in this study. A combination of agonists (albeit at a single concentration of each) give more than an additive response when compared to the responses to the individual agonists. However, the error bars are high and it is not clear how many cells were examined and from how many animals (this should be in Figure legend). In the figure legend the number of (n) independent experiments not provided.*

The requested information has been added in the figure and figure legend.

*5) Also a dose response curve of either LY34540 (or LSP4-2022) in the presence of a low dose of LSP4-2022 (or LY34540) would have strengthened this crucial point.*

It is difficult to perform a dose response curve in a slice preparation. Instead of doing these experiments, we preferred to generate data from slices prepared from mGlu4 KO mice (see revised Figure 7).

*6) Figure 1—figure supplement 1 what do the different colors in the plots mean. How do they help differentiate homomeric from heteromeric signals?*

The colors are related to the FRET intensity (color scale, the higher FRET is red, while the lower is dark blue). We want to make clear that the three graphs were generated from the same set of cells transfected with various amount of CT-mGlu2 and SNAP-mGlu4. The TOP graph reveals FRET data after CLIP labeling with both donor and acceptor substrates. It shows the best conditions leading to high amount of mGlu2 homodimers. The middle graph reports data after labeling the cells with donor and acceptor SNAP substrates. It shows the best conditions to get higher amount of mGlu4 homodimers. The bottom graph shows data after labeling the cells with a SNAP donor substrate, and a CLIP acceptor substrate. Then this bottom graph shows the best condition to get higher proportion of mGlu2-4 heterodimers. By comparing the three graphs, one can identify the conditions that favor the heteromer expression relative to each of the homodimers. The figure legend has been modified.

*7.)There is a mismatch between the title of the figure and figure legend – the authors are requested to check this carefully. For example, legend on*
Figure 1—figure supplement 1
*says mGlu4 but title of figure says mGlu2, legend for*
Figure 1—figure supplement 1
*says mGlu2 but title of figure says mGlu4. Similarly in*
Figure 1—figure supplement 2
*title in the figure says mGlu2 agonist but Results section states mGlu4 agonist.*

We thank the referee for pointing these mistakes. These have been corrected.

*8) Most importantly, neither the co-localization of the two receptors in this system nor the absence of synergy in animals lacking one of the receptors has been explored.*

Experiments have been performed in slices prepared from mGlu4 KO mice. The data are now included in the revised Figure 7. Data show that the effect of LSP4-2022 is lost in these slices indicating that mGlu4 was involved. In contrast, the effect of the LY354740 remains unaffected. We also found that LSP4-2022 no longer potentiate the effect of LY354740 in the absence of mGlu4.

Regarding the co-localization studies, it is known that both mGlu2 and mGlu4 are expressed in these terminals, and our data confirm this since both LY354740 and LSP4-2022 inhibit transmission. Performing co-localization studies would need more time for us to conduct such studies especially because these would need electron microscopy to really be meaningful. We thought that such studies are out of the scope of the present study, concentrating in original approaches to specifically study the pharmacological properties of mGlu heterodimers. We also preferred to concentrate on the time available to produce a revised manuscript, on the characterization of mGlu4 KO slices.

*Reviewer #3:*

*This manuscript presents very nice data in support of mGlu2-mGlu4 heterodimer receptors. The FRET experiments are convincing and demonstrate that the pharmacology of herodimers can be distinguished from that of either homodimers. Second, the authors show that several compounds have distinct actions on hererodimers compared to homodimers, such as different Hill coefficients or partial agonist or synergistic activities. Finally the experiments measuring cell swelling of cultured striatal neuron cell line as a proxy for adenylate cyclase activation are inovative and reveal the synergistic activity of BINA and VU0415374 in neurons. My only question about this manuscript is that the evidence of mGlu2-4 heterodimers in the lateral perforant path is most suggestive, but it is not conclusive. The synergistic effects shown could be caused downstream of the receptors in the Gi-coupled pathways. Given the pharmacological fingerprints presented in Figure 4 or 6, can similar experiments be done in hippocampal slices?*

The referee is right, one cannot exclude the possibility of a synergistic effect between two Gi coupled receptors, although such a synergy has not been reported so far. We therefore modified our manuscript to minimize our conclusion of the existence of mGlu2-4 heterodimers in the perforant path terminals, and rather claim that our data are consistent with the presence of such heterodimers. In order to offer a stronger conclusion, we aimed at studying, in recombinant systems, the possible synergistic effects between mGlu2 and mGlu4 homodimers. Although no such synergy could be detected with the TR-FRET biosensors (the mGlu2 pharmacological properties shown in Figure 1 were measured in the presence of mGlu4 and mGlu2-4), such observation did not exclude a synergistic action downstream of receptor activation.

We then aimed at expressing mGlu2 and mGlu4 homodimers in cells without the presence of mGlu2-4 heterodimers. This was theoretically possible using mGlu2C1, mGlu2-C2 and mGlu4. We were expecting that co-transfecting these three constructs would lead to mGlu2 and mGlu4 homodimers at the cell surface only, since neither mGlu2-C1-mGlu4 nor mGlu2-C2-mGlu4 heterodimers were expected to reach the cell surface due to the presence of the ER retention signal in the C1 and C2 tails. Unfortunately, these two combinations were found in sufficient quantities at the cell surface, preventing any analysis of a possible synergistic effect between mGlu2 and mGlu4 homodimers. We then co-expressed mGlu2 and the Gi-coupled δ opioid receptor (DOR), and examined whether activation of DOR could potentiate the effect of the mGlu2 agonist in inhibiting Forskoline-induced cAMP formation using the pGlo-sensor. As show in Figure 6—figure supplement 2, no potentiation was found, despite the equivalent expression of both receptors, and the correct coupling of both receptors to the inhibition of cAMP formation.

Although not fully conclusive, we think these additional data strengthen our view.

Regarding the last comments of the referee, though we agree that such experiments would be great, it is quite difficult, time consuming, and expensive in terms of the amount of drug needed to perform a full dose-response curve using electrophysiological recordings in brain slices. We then preferred to concentrate our work in getting data with the mGlu4 KO mice.